# Quantifying the impact of global nitrate aerosol on tropospheric composition fields and its production from lightning NO$_x$

Ashok K. Luhar[1], Anthony C. Jones[2], Jonathan M. Wilkinson[2,a]

[1]CSIRO Environment, Aspendale, Victoria 3195, Australia
[2]Met Office, Fitzroy Road, Exeter, EX1 3PB, UK
[a]now at: Forecast Department, European Centre for Medium-Range Weather Forecasts, Reading, UK

*Correspondence to*: Ashok K. Luhar (ashok.luhar@csiro.au)

**Abstract.** Several global modelling studies have explored the effects of lightning-generated nitrogen oxides (LNO$_x$) on gas-phase chemistry and atmospheric radiative transfer, but few have quantified LNO$_x$'s impact on aerosol, particularly when nitrate aerosol is included. This study addresses two key questions: 1) how does including nitrate aerosol affect properties such as tropospheric composition, and 2) how do these effects depend on lightning parameterisation and LNO$_x$ levels? Using the Met Office's Unified Model–UK Chemistry and Aerosol (UM-UKCA) global chemistry-climate model, which now includes a modal nitrate aerosol scheme, we investigate these effects with two lightning flash-rate parameterisations. Our findings show that both nitrate aerosol and LNO$_x$ significantly impact tropospheric composition and aerosol responses. Including nitrate aerosol reduces global mean tropospheric OH by 5%, decreases the tropospheric ozone burden by 4–5%, increases methane lifetime by a similar amount, and alters the top-of-atmosphere (TOA) net downward radiative flux by –0.4 W m$^{-2}$. The inclusion of nitrate also shifts the aerosol size distribution, particularly in Aitken and accumulation modes. A 5.2 Tg[N] yr$^{-1}$ increase in LNO$_x$ from a zero baseline results in global aerosol increases of 2.8% in NH$_4$, 4.7% in fine NO$_3$, 12% in coarse NO$_3$, and 5.8% in SO$_4$ mass burdens. This much LNO$_x$ increase causes relatively small positive changes in aerosol optical depth, TOA radiative flux, and cloud droplet number concentration compared to when nitrate is included. The results, based on a fast uptake rate for HNO$_3$ to produce NH$_4$NO$_3$, likely represent an upper limit on nitrate effects.

# 1 Introduction

Liquid or solid particles in the atmosphere, called aerosols, have a considerable impact on Earth's energy budget and thus the climate system, and also impact air quality. Aerosols can be emitted directly from both anthropogenic and natural sources (termed primary aerosols), or they can be chemically produced in the atmosphere from precursor gases by condensation of vapours on pre-existing particles or by nucleation of new particles (termed secondary aerosols). Although atmospheric aerosol can comprise many different species, the main groups include sulfate ($SO_4^{2-}$, or simply $SO_4$), nitrate ($NO_3^-$, or simply $NO_3$), ammonium ($NH_4^+$, or simply $NH_4$), carbonaceous aerosols (e.g., organic carbon and black carbon), sea salt, and mineral dust, with their sizes varying from nanometres to tens of micrometres dependent on the formation process (Szopa et al., 2021). Apart from exerting a direct radiative forcing (i.e. a change in the energy budget at the top-of-atmosphere (TOA)), aerosols also impact clouds through aerosol–cloud interactions which in turn impact radiative forcing indirectly.

Sulfate and nitrate are predominantly secondary inorganic aerosols produced from atmospheric oxidation and are often the major components of fine mode aerosol (Haywood and Boucher, 2000). They are strong scatterers of the incoming solar energy (the 'aerosol direct effect') and thus exert a negative radiative forcing on climate (meaning a cooling of the atmosphere). These aerosols also act as the cloud condensation nuclei (CCN), and modify cloud properties, such as cloud albedo, persistence or lifetime of clouds and precipitation rate (the 'aerosol indirect effect').

Sulfate is formed by oxidation of precursor sulfur dioxide ($SO_2$) in gas phase and in aqueous phase in cloud droplets. The main precursor species for the chemical formation of nitrate aerosol are reactive gases of nitric acid ($HNO_3$) and ammonia ($NH_3$). $HNO_3$ is the oxidation product of nitrogen oxides ($NO_x$). $NO_x$ is a mixture of nitric oxide (NO) and nitrogen dioxide ($NO_2$), and it is primarily emitted from anthropogenic burning of fossil fuel and from natural sources, such as biomass burning, soil emissions, and lightning. It is also an important air pollutant by itself, and further impacts air pollution and radiation, for example, through net tropospheric ozone ($O_3$) production and changes in tropospheric methane lifetime caused by changes in the hydroxyl radical (OH). Major emissions of $NH_3$ gas include agricultural sources, which include volatilisation of livestock manure and mineral fertiliser application. Sulfate aerosol also indirectly influences nitrate aerosol and $O_3$ levels through changes in oxidation rates and by eroding ammonia concentrations.

Sulfate and nitrate particles (or in particles) can be dry or in aqueous solution and their production takes place through complex chemical pathways (Finlayson-Pitts and Pitts, 2000). In short, $SO_2$ and $NO_x$ are oxidised into sulfuric acid ($H_2SO_4$) (liquid aerosol droplets) and $HNO_3$ (atmospheric gas), respectively. $H_2SO_4$ reacts with $NH_3$ to produce aerosol of ammonium sulfate ($NH_4HSO_4$ and $(NH_4)_2SO_4$). Thus, tropospheric sulfate aerosol may be considered as consisting of sulfuric acid particles that are partially or totally neutralised by $NH_3$. After $H_2SO_4$ is neutralized, any excess $NH_3$ then combines with $HNO_3$ to form aerosol of ammonium nitrate ($NH_4NO_3$). Low temperature, high relative humidity, and elevated fine particulate matter favour nitrate production (Szopa et al., 2021; references therein).

Ammonium and nitrate aerosols formed through these gas-to-particle reactions are a large portion of fine-mode particles (with a diameter < 1μm) affecting both climate and air quality, particularly over populous regions in the Northern Hemisphere (Szopa et al., 2021). In addition to fine mode, coarse-mode nitrate aerosol is formed when $HNO_3$ condenses irreversibly onto existing sea-salt and dust aerosols to produce sodium nitrate ($NaNO_3$) and calcium nitrate ($Ca(NO_3)_2$) salts, respectively (Li and Shao, 2009). The coarse-mode nitrate aerosol dominates the global mass burden of nitrate, which may be important from air quality side, but has little radiative effect in the solar spectrum compared to the fine-mode nitrate (Bian et al., 2017; Hauglustaine et al., 2014).

Apart from its direct effects on climate and air quality, nitrate aerosol through its deposition also plays a part in constraining net primary productivity, thus altering sequestration of carbon and ecological effects (Bian et al., 2017). Nitrate aerosol is expected to become even more important in the future atmosphere due to the continued increase in nitrate precursor emissions (e.g., $NH_3$ and $NO_x$) and the decline in $SO_2$ due to stricter emissions regulations (Bellouin et al., 2011; Hauglustaine et al., 2014). For example, this could be due to increasing use of ammonia based fertilisers and the potential use of ammonia as an effective medium for storing and transporting hydrogen as a fuel in a competitive net zero carbon economy. Although nitrate aerosol has been included in some global models, such as the chemical transport model GEOS-Chem (e.g., Park et al., 2004) and the chemistry-climate models GISS (e.g., Bauer et al., 2007) and GFDL (e.g., Paulot et al., 2016), it is often ignored in global chemistry-climate models (Tost, 2017). This may be partly due to the computational cost of simulating nitrate, combined with the chemical complexity of its formation and the semi-volatile nature of ammonium nitrate, which can reevaporate into the atmosphere (e.g., Stelson et al., 1979). In fact, out of the ten global Earth system models with atmospheric chemistry that participated in the Aerosol and Chemistry Model Intercomparison Project (AerChemMIP) under the Coupled Model Intercomparison Project Phase 6 (CMIP6), which aims to assess the effects of reactive gases and aerosols on Earth's climate, only the GISS and GFDL models explicitly treated nitrate aerosol along with an interactive stratospheric and tropospheric chemistry scheme (Thornhill et al., 2021).

As mentioned above, lightning is a major source of $NO_x$, particularly in the tropical to subtropical middle to upper troposphere where lightning is mostly discharged (Murray, 2016; Bucsela et al., 2019). Although lightning-generated $NO_x$ (abbreviated as $LNO_x$) constitutes only about 10% of the total $NO_x$ source emissions globally, it has an inordinately large effect on tropospheric composition (e.g., Murray, 2016; Luhar et al., 2021), such as the OH and $O_3$ mixing ratios. To give an example, whilst $NO_x$ emissions from lightning are comparable in magnitude to those from soils or biomass burning, they contribute about three times as much to the total tropospheric $O_3$ column (Dahlmann et al., 2011). This is because, in the middle to upper troposphere where lightning $NO_x$ is released, the $O_3$ production efficiency per unit of $NO_x$ is significantly higher (~ 100 molecules of $O_3$ per molecule of $NO_x$) compared to near the surface (~ 10–30 molecules of $O_3$ per molecule of $NO_x$) due to the higher amount of UV radiance, lower concentrations and longer lifetimes of $NO_x$ (days instead of hours), and cooler temperatures affecting ozone loss chemistry at such altitudes (Dahlmann et al., 2011).

Previous global modelling studies have demonstrated the importance of LNO$_x$ on atmospheric gas-phase chemistry and oxidation capacity (see for example, Labrador et al., 2005, Schumann and Huntrieser, 2007; Finney et al., 2016; Gressent et al., 2016; Gordillo-Vázquez et al., 2019, and Luhar et al., 2021), and also its impact on cloud cover and atmospheric radiative transfer (e.g., Luhar et al., 2022). Even though these modelling studies did not explicitly include nitrate aerosol processes, an indirect impact of LNO$_x$ on aerosol is implicit through perturbations to the oxidation capacity of the

atmosphere (e.g. via changes in OH and O$_3$), for example enhancement of new particle formation and aerosol abundances stemming from faster oxidation rates of gas-phase sulfur to sulfate (Murray, 2016; Tost, 2017; Luhar et al., 2022).

The area of quantifying the role of LNO$_x$ production on aerosol, particularly with nitrate aerosol included, has only received very limited attention compared to its role on gaseous atmospheric composition, and this could be due to reasons such as the inference that the low LNO$_x$ emission ($\sim$ 12%) compared to anthropogenic and biomass burning NO$_x$ sources contributes

negligibly to nitrate concentrations. To our knowledge, the global modelling by Tost (2017), which involves a modal aerosol scheme with nitrate included, is the only study to explicitly examine the impact of LNO$_x$ on aerosol. It shows that LNO$_x$ (parameterised via the Price and Rind (1992) (PR92) scheme described below) is a significant source of nitrate in the upper troposphere and influences the aerosol size distribution and radiation. It is reported that chemical conversion of LNO$_x$ into HNO$_3$ is more favourable in the middle to upper troposphere, where lightning NO$_x$ mostly occurs, as compared to within the

atmospheric boundary layer (where the dominant NO$_x$ and NH$_3$ sources are located) due to differences in chemical composition, chemical reactivity, and loss processes (Tost, 2017). Tost (2017) points to observational support for the occurrence of both NH$_3$ and NO$_3$ aerosol in convective outflows so that the formation of NH$_4$NO$_3$ is likely because of the low temperatures in the upper troposphere. Therefore, LNO$_x$ can change the spatial distribution of nitrate concentrations and concomitant climate impacts. Given the emerging importance of nitrate as sulfate concentrations wane it is important to

assess the relative importance of all nitrate sources. At the same time, how modelled global atmospheric composition is impacted when nitrate is accounted needs to be quantified.

A more comprehensive study involving the sensitivity of tropospheric composition (including aerosol) to global lightning parameterisation with and without nitrate is currently lacking and the present paper aims to address that. Recently, a nitrate scheme has been included in the modal aerosol scheme of the Met Office's Unified Model (UM) – United Kingdom

Chemistry and Aerosol (UKCA) global chemistry-climate model, hereafter denoted UM-GA8.0-UKCA or UM-UKCA, where GA8.0 refers to the configuration of the Global Atmosphere in the UM (Jones et al., 2021). The aerosol scheme in UM-UKCA is a prognostic double-moment scheme (Mann et al., 2010) which transports both particle number and mass concentrations in size classes. In this paper, we apply UM-UKCA to study the sensitivity of tropospheric composition and radiation to the lightning parameterisation with and without nitrate aerosol included.

Ultimately, the questions we seek to answer using UM-UKCA are: 1) what is the impact of adding nitrate to UM-UKCA on tropospheric composition fields (such as O$_3$, OH, NO$_x$, methane lifetime, and aerosol), AOD and radiation, and 2) what is the dependency of these impacts on the lightning parameterisation in the model? For this purpose, we apply UM-UKCA without

and with nitrate and vary LNO$_x$ through the use two empirical parameterisations of lightning flash rate: (a) the PR92 parameterisations for ocean and land which are a function of convective cloud-top height and which most global chemistry-climate models use (Tost, 2017; Archibald et al., 2020), and (b) Luhar et al.'s (2021) (termed Lu21) flash-rate parameterisations which improve upon the PR92 parameterisation for the ocean. Simulations are also conducted with no LNO$_x$. Our modelling can be considered as a study of sensitivity of global fields of interest to changes in lightning NO$_x$ without and with nitrate aerosol. An outline of the paper is as follows. In Section 2, we provide details of the LNO$_x$ parameterisations used; Section 3 briefly describes the global UM-UKCA model configuration used, with some extra detail about model's aerosol scheme with nitrate and the model simulation setup; this is followed by results and discussion in Section 4 and then conclusions in Section 5.

## 2 Lightning-generated NO$_x$ parameterisations

In traditional lightning parameterisations used in general circulation models, the amount of LNO$_x$ is estimated as

$$LNO_x = P_{NO} \times F, \tag{1}$$

where $P_{NO}$ is the amount of NO produced per flash. The flash rate $F$ is computed for every model time step and each grid column.

In global chemical transport and chemistry-climate models (including most CMIP6 models), $F$ (flashes per minute) is most commonly parameterised in terms of convective cloud-top height using Price and Rind's (1992) (PR92) formulas for land ($F_L$) and ocean ($F_O$) as follows:

$$F_L = 3.44 \times 10^{-5} H^{4.9}, \tag{2}$$

$$F_O = 6.4 \times 10^{-4} H^{1.73}, \tag{3}$$

where the convective cloud-top height (H, in km) is diagnosed on a time-step basis from UM's convection scheme. and a minimum cloud depth of 5 km is required for LNO$_x$ emissions to be activated. A spatial calibration factor is applied to adjust the above flash rate formulas for given horizontal model resolutions (Price and Rind, 1994; Luhar et al., 2021). The oceanic flash rates obtained from Eq. (3) are about 2 to 3 orders of magnitude smaller as compared to those computed using Eq. (2) for overland clouds.

The calculated flash rate ($F_L$ or $F_O$) is divided into intracloud (IC) and cloud-to-ground (CG) components, $F_{IC}$ and $F_{CG}$, and respective emission factors for the amount of NO released per IC and CG flash, $P_{NO,IC}$ and $P_{NO,CG}$, are applied, i.e.,

$$LNO_x = P_{NO,IC} \times F_{IC} + P_{NO,CG} \times F_{CG}. \tag{4}$$

The fraction of CG lightning flashes is determined based on cold cloud thickness, following an empirical relationship developed by Price and Rind (1993), where cold cloud thickness is further parameterised as a function of latitude. The remaining fraction is then equal to the IC flash fraction. These fractions multiplied with the calculated flash rate ($F_L$ or $F_O$) give $F_{CG}$ and $F_{IC}$, respectively. The calculated NO at a specific location and time step is distributed vertically in the grid column using a linear distribution in log(pressure) coordinates. For IC flashes, this extends from 500 hPa to the cloud top, and for CG flashes, from 500 hPa to the surface (Archibald et al., 2020; Luhar et al., 2021). Eq. (3) is known to greatly underpredict flash rates over the ocean. Luhar et al. (2021) tested the PR92 flash-rate formulas in the Australian Community Climate and Earth System Simulator – UKCA (or ACCESS-UKCA) global chemistry-climate model without nitrate aerosol (which is essentially UM-UKCA with the UM at vn8.4) using satellite based lightning data and concluded that whilst the PR92 formula for land (Eq. (2)) performs satisfactorily, the oceanic formula, Eq. (3), predicts a mean global flash rate that is smaller by a factor of approximately 30 compared to the observed (a predicted global oceanic average of 0.33 flashes s$^{-1}$ compared to the observed 9.16 flashes s$^{-1}$), thus results in a proportional underestimation of LNO$_x$ over the ocean.

Luhar et al. (2021) formulated the following updated flash-rate parameterisations following the scaling relationships between thunderstorm electrical generator power and storm geometry developed by Boccippio (2002), coupled with available data:

$$F_L = 2.40 \times 10^{-5} H^{5.09}, \tag{5}$$

$$F_O = 2.0 \times 10^{-5} H^{4.38}. \tag{6}$$

In the study by Luhar et al. (2021), Eq. (5) did very similar to Eq. (2) in estimating the spatial pattern of the global continental lightning flash rate, giving a global mean value of 35.9 flashes s$^{-1}$ compared to the satellite climatological value of 38.5 flashes s$^{-1}$. The updated oceanic parameterisation (6) simulated the oceanic (and thus the global total $F = F_L + F_O$) flash-rate data much better, giving a mean global oceanic flash rate of 9.1 flashes s$^{-1}$ compared to 7.7 flashes s$^{-1}$ derived from the satellite climatology. However, there was still an underestimation of flash frequencies in the extratropics, which is probably an inherent limitation of the convective cloud-top height approach to parameterise $F$. The updated flash-rate parameterisations resulted in an increase in the global LNO$_x$ by ~ 40% (from the default value of 4.8 Tg[N] yr$^{-1}$), causing a significant change on the tropospheric composition compared to the default model run.

There is a large uncertainty in the global LNO$_x$ amounts. For example, Schumann and Huntrieser (2007) cite an uncertainty of 2–8 Tg[N] yr$^{-1}$. Other estimates include 4–8 Tg[N] yr$^{-1}$ (Martin et al., 2007), and ~ 9.0 Tg[N] yr$^{-1}$ (Nault et al., 2017). The range of global LNO$_x$ in five CMIP6 Earth system models for the present-day conditions ranged 3.2–7.6 Tg[N] yr$^{-1}$ (Griffiths et al., 2021; Szopa et al., 2021). A major reason for this is due to the poor constraining of the quantity of NO produced per flash. The default in the UM-UKCA model, as used by Luhar et al. (2021), is $P_{NO,IC} = P_{NO,CG} = P_{NO} = S_f \times 10^{26}$ molecules of NO produced per flash, where it is assumed that the scaling factor $S_f = 2$. This is equal to $P_{NO} = 330$ moles NO produced per flash, which is close to the middle of the large range 70–665 moles NO per flash reported in the scientific literature based

on observations (Luhar et al., 2021). Here, we use $S_f = 1.7$, so $P_{NO}$ = 280 moles NO per flash, which can be compared with 280 $\pm$ 80 moles NO produced per flash obtained by Marais et al. (2018) using satellite-based lightning data and the Ozone Monitoring Instrument (OMI) $NO_2$ columns coupled with the GEOS-Chem global chemical transport model, and 310 moles NO per flash suggested by Miyazaki et al. (2014) based on an assimilation into a global CTM of satellite observations of atmospheric composition and flash distribution.

Because $P_{NO}$ is taken independent IC and CG in our model, the partitioning of flash rate into the IC and CG flash rates only affect the shape of the vertical profile of $LNO_x$ distribution. The total amount of $LNO_x$ released remains independent of this partitioning.

### 3 The Met Office Unified Model (UM) with global chemistry and aerosol

We use the Met Office Unified Model (UM) which includes global atmosphere and coupled modelling systems from weather
to climate timescales. The latest available release (at the time) of the UM (vn13.2) involving the science configurations Global Atmosphere vn8.0 (GA8.0) and Global Land vn9.0 (GL9.0) configuration of the Joint UK Land Environment Simulator (JULES) land surface model was selected. (The new nitrate aerosol scheme was included in the UM at UM vn11.8 (Jones et al., 2021) with the corresponding science configurations GA7.1 and GL7.0 as described by Walters et al. (2019) and updates between GA7.0 and GA8.0 pertinent to aerosols described in Jones et al. (2022)).

The horizontal resolution of the model is 1.875° longitude × 1.25° latitude and there are 85 staggered hybrid-height levels which extend from the surface to 85 km in the vertical (the so-called N96L85 climate configuration). The resolution in the vertical becomes coarser, with the lowest 50 levels under 18 km altitude. The model was run in atmosphere-only mode with a dynamical timestep of 20 min. Monthly climatologies of sea surface temperature and sea ice fields for the year 2000 are prescribed, which are generated by averaging over the 1995–2004 time series data created for CMIP6 atmosphere-only
model simulations. This is the same model setup as Jones et al (2021), albeit using the GA8.0 rather than GA7.1 atmosphere model.

Atmospheric composition in the UM is described by the United Kingdom Aerosol and Chemistry (UKCA) model (https://www.ukca.ac.uk), with model's chemical solver called every 60 min. The UKCA configuration used here employs a combined stratosphere–troposphere chemistry scheme (namely, StratTrop1.0) (Archibald et al., 2020), which also contains
200 the multi-modal, multi-component, double-moment GLObal Model of Aerosol Processes (GLOMAP)-mode aerosol microphysics scheme as described by Mann et al. (2010) and Mulcahy et al. (2020). GLOMAP-mode as coupled to UKCA chemistry is termed UKCA-mode.

With 84 species and 291 chemical reactions, the StratTrop1.0 scheme in UKCA simulates the chemical cycles of $O_x$, $HO_x$, $NO_x$, and halogenic compounds; the oxidation of carbon monoxide (CO), methane ($CH_4$) and other volatile organic

compounds (VOCs); and heterogenous chemistry on Polar Stratospheric Cloud and tropospheric aerosols. The Fast-JX scheme (see Telford et al., 2013) is used to model interactive photolysis. Gaseous wet and dry deposition are included.

Atmospheric radiative transfer is described via a two-stream approximation, with 9 longwave bands and 6 shortwave bands (Manners et al., 2023). The radiation changes include both direct aerosol radiative forcing and indirect radiative effects of clouds and atmospheric composition changes. The Prognostic Cloud fraction and Prognostic Condensate (PC2) scheme is used to model large-scale cloud (Wilson et al., 2008) and cloud microphysics is simulated using a single-moment scheme (Wilson and Ballard, 1999) with extensive revisions (as included in GA8.0). The convection scheme is based on a mass flux-approach (Gregory and Rowntree, 1990) with several extensions to include convective momentum transport and downdrafts (Walters et al., 2019).

The full modelling system we use is termed UM-GA8.0GL9.0-UKCA, which in this paper is referred to as UM-UKCA.

### 3.1 UKCA-mode aerosol scheme with nitrate

UKCA-mode as used here is a double-moment aerosol microphysics scheme that transports aerosol number and mass concentrations in lognormal size modes (four soluble and one insoluble) (Mann et al., 2010; Mulcahy et al., 2020). This is the default UKCA-mode setup 2 (see Table 1). For each mode, the median aerosol dry radius is allowed to evolve within specified size ranges, but the lognormal standard deviation (i.e. "mode width") is kept fixed. This scheme thus resolves the differential growth of particles and their composition across the aerosol size range including internal mixtures. The default UKCA-mode configuration involves sulfate ($SO_4$), organic matter (OM), black carbon (BC) and sea salt (SS). Species in each mode are considered as an internal mixture. On the other hand, mineral dust is treated outside of UKCA-mode and is represented by a six-bin scheme developed as part of the older single-moment Coupled Large-scale Aerosol Simulator for Studies in Climate (CLASSIC) framework (Woodward et al., 2001; Bellouin et al., 2013).

With the new nitrate scheme (Jones et al., 2021), which follows Hauglustaine et al. (2014) and Rémy et al. (2019), ammonium ($NH_4$), nitrate ($NO_3$), and coarse nitrate (denoted coarse$NO_3$) are added to the standard aerosols, namely $SO_4$, OM, BC and SS, and the new UKCA-mode setup consists of 28 aerosol tracers (23 mass + 5 number concentrations) in total (Table 1) (this UKCA-mode setup together with the CLASSIC dust scheme is referred to as setup 10). Note that here "$NO_3$" refers exclusively to semi-volatile $NO_3$ linked with $NH_4$, whereas "coarse$NO_3$" pertains to stable $NO_3$ related to sea salt and dust. $NH_4$ and $NO_3$ mass is introduced into the Aitken soluble and accumulation soluble modes and it may be moved to the coarse soluble mode through coagulation and mode merging, whereas coarse$NO_3$ is restricted to the accumulation soluble and coarse soluble modes. Only nitrate condensing on bins 2-6 (with bin 2 and half of bin 3 mapped to the accumulation mode and the other half of bin 3 and bins 4, 5, and 6 mapped to the coarse mode) of the CLASSIC mineral dust scheme is considered. The nitrate scheme requires $NH_3$ and $HNO_3$ to be available (emission of $NH_3$ is prescribed, while gaseous $HNO_3$ is produced by various chemical reactions in the atmosphere). Full detail of the methodology is reported by Jones et al. (2021).

The component of the new nitrate scheme dealing with the formation of fine-mode $NH_4NO_3$ from the condensation of $HNO_3$ and $NH_3$ is numerically solved first, prior to the condensation of $HNO_3$ on coarse aerosols (i.e., dust and sea salt). Most fine-mode nitrate schemes assume that $NH_4NO_3$ concentrations reach thermodynamic equilibrium instantaneously, without accounting for the kinetic limitations on the condensation of $HNO_3$ or $NH_3$ onto existing aerosol particles. Instead, our quasi-instantaneous thermodynamic equilibrium scheme assumes an exponential decay of the gas phase toward equilibrium, using an equilibration time scale ($\tau_e$). This approach is based on Schwartz's (1986) first-order uptake theory and incorporates correction factors from Fuchs and Sutugin (1970) to account for molecular effects and limitations in interfacial mass transport. $\tau_e$ is a function of the $HNO_3$ condensation or uptake rate coefficient ($\gamma$), a key parameter in the first-order uptake theory and defined as the number of gas molecules condensing on a particle divided by the number impacting onto the particle surface. The higher the uptake coefficient the smaller the equilibration time scale. The benefit of using such a scheme is that it realistically constrains the rate at which $NH_4NO_3$ concentrations achieve equilibrium. .

Jones et al. (2021) tested the sensitivity of $NH_4NO_3$ aerosol concentrations to the $HNO_3$ uptake coefficient for the $NH_3$-$HNO_3$ uptake on Aitken and accumulation soluble particles (Table 1) with two values selected from the literature, $\gamma = 0.193$ (FAST) and 0.001 (SLOW), representing fast and slow uptake rates, respectively. They found that, generally, the fast uptake value shows a higher spatial correlation with measured nitrate surface concentrations whereas the slow value simulates their magnitudes better. They also found that compared to FAST, the SLOW value led to a 58% and 52% reduction in the global near-surface concentration and burden of fine particulate nitrate, respectively. The reductions in $NH_4$ were 24% and 15%, while coarse mode $NO_3$ remained almost unchanged. Aerosol optical depth decreased by 6%, and the magnitude of the TOA net downward radiative flux changed by 63%. This sensitivity test showed that despite a two-hundredfold variation in the uptake rate, the model's response was nonlinear and perhaps less sensitive than expected. In this study, we use the FAST value $\gamma = 0.193$, which is currently the default in UKCA-mode. Jones et al (2021) showed that this value produces similar results globally to the widely utilised assumption of instantaneous thermodynamic equilibrium. This suggests that our results likely represent the upper end of efficiency of $NH_4$ and $NO_3$ production and its impact in the UM. Jones et al. (2021) recognised that rather than being globally invariant, $\gamma$ may vary with aerosol composition, temperature, and relativity humidity, and needs better constraining, thus needing further research and future model development outside the scope of the present study.

In our nitrate scheme, coarse nitrate is present in the accumulation and coarse soluble modes. Following $NH_4NO_3$ production and the associated update to $HNO_3$ concentrations, the first-order uptake parameterisation is further employed to model the irreversible uptake of $HNO_3$ on sea salt and dust to produce $NaNO_3$ and $Ca(NO_3)_2$, respectively. This reaction is slower than ammonium nitrate production, therefore numerically ammonium nitrate production is solved first. The $HNO_3$ uptake coefficients for CLASSIC dust and sea salt are relative humidity dependent variables based on measurements from Fairlie et

al. (2010) and Sander et al. (2011), respectively. Dust is assumed to uniformly constitute 5% $Ca^{2+}$ by mass (Jones et al., 2021).

The model's radiation scheme and treatment of hygroscopic growth has been modified to include the direct radiative effects of nitrate aerosol and the indirect impacts on clouds, enabling a comprehensive nitrate impacts assessment in future UM studies (Jones et al., 2021).

**Table 1: Aerosol size distribution used in the UKCA-mode aerosol scheme. Species are sulfate ($SO_4$), black carbon (BC), organic matter (OM), sea salt (SS), ammonium ($NH_4$), nitrate ($NO_3$), and coarse nitrate (coarse$NO_3$) (adapted from Jones et al. (2021)).**

| Aerosol mode | Geometric mean diameter (nm) | Geometric standard deviation | Species in the standard aerosol scheme | Additional species with nitrate scheme |
|---|---|---|---|---|
| Nucleation soluble | 1–10 | 1.59 | $SO_4$, OM | - |
| Aitken soluble | 10–100 | 1.59 | $SO_4$, BC, OM | $NH_4$, $NO_3$ |
| Accumulation soluble | 100–1000 | 1.4 | $SO_4$, BC, OM, SS | $NH_4$, $NO_3$, coarse$NO_3$ |
| Coarse soluble | > 1000 | 2.0 | $SO_4$, BC, OM, SS | $NH_4$, $NO_3$, coarse$NO_3$ |
| Aitken insoluble | 10–100 | 1.59 | BC, OM | - |

## 3.2 Model simulation setup

We conducted six UM-UKCA simulations for various combinations of the $LNO_x$ and nitrate schemes, including no nitrate and no $LNO_x$ cases (Table 2). The model experimental setup is very similar to that by Jones et al. (2021). All simulations are

initialised with results from a previously spun-up model experiment and are run for model years (1989–2008). To further eliminate any spin-up effects, model output from only the last 15 years (1994-2008) is used for analysis. The model is forced by prescribed sea-surface temperature and sea-ice fields specified as monthly varying climatologies for the 'year 2000', which are essentially averages over the 1995–2004 timeseries data developed for CMIP6 atmosphere-only simulations. No meteorological nudging is used so as to allow feedback mechanisms to create an individual climate state. The use of

perpetual year 2000 conditions in our simulation design follows standard simulation protocol for the development of the UKCA model in the Met Office. For a statistical robustness of the included feedbacks in the model, instead of examining individual years of the simulations, we present results that are averages over the 15-yr simulation period.

The gaseous and aerosol emissions are mainly prescribed from the CMIP6 historical emissions inventory as monthly fields and they include anthropogenic, biomass burning and natural categories. Jones et al. (2021) provide further details on these

emissions including global total for each UKCA chemical species. In our simulations, monthly emissions for the year 2000, which are generated by averaging over the 1995–2004 timeseries data, are used and the emissions are computed every model

time step of 20 min. The only extra species emission used by the model when the nitrate scheme is invoked is that of $NH_3$, whose global total was 53.6 Tg[N] yr$^{-1}$ (Table 3).

The global total $NO_x$ emission (excluding $LNO_x$) was 49.5 Tg[N] yr$^{-1}$ (Table 3), of which the soil, anthropogenic surface, biomass burning, and aircraft components were 11.1, 72.3, 15.2 and 1.4%, respectively. Note that the $LNO_x$ emission is not included in the input emissions database and is generated interactively as described in Section 2.

Concentrations of long-lived species $CO_2$, $CH_4$, $N_2O$ and $O_3$ depleting substances are prescribed as lower boundary conditions at the surface (Archibald et al., 2020). All simulations used the optional two-layered oceanic process-based scheme proposed by Luhar et al. (2018) (configuration corresponding to the Ranking 1 in their Table one) for dry deposition ozone to the ocean.

**Table 2: Details of six UM-UKCA simulations performed in this study. The indicated lightning schemes are Price and Rind (1992) (PR92) and Luhar et al. (2021) (Lu21). The uncertainty corresponds to a 1-sigma standard deviation calculated from annual means over 15 years of simulation after linear detrending (sample size = 15).**

| Simulation name | Nitrate status | Lightning scheme | Global $LNO_x$ (Tg[N] yr$^{-1}$) | Tropospheric methane lifetime (yr) | Tropospheric $O_3$ burden (Tg) |
|---|---|---|---|---|---|
| NN_0 | No nitrate | - | 0 | $8.55 \pm 0.03$ | $278.5 \pm 2.2$ |
| NN_PR | No nitrate | PR92 | 3.41 | $7.49 \pm 0.03$ | $325.2 \pm 3.5$ |
| NN_Lu | No nitrate | Lu21 | 5.24 | $7.03 \pm 0.02$ | $348.8 \pm 3.0$ |
| WN_0 | With nitrate | - | 0 | $9.15 \pm 0.04$ | $260.8 \pm 3.4$ |
| WN_PR | With nitrate | PR92 | 3.36 | $7.90 \pm 0.03$ | $307.1 \pm 2.3$ |
| WN_Lu | With nitrate | Lu21 | 5.18 | $7.38 \pm 0.02$ | $332.6 \pm 2.5$ |

**Table 3: Annual totals of global emissions of nitrogen types prescribed in the UM-UKCA simulations.**

| Species | Source | Emissions (Tg[N] yr$^{-1}$) |
|---|---|---|
| $NH_3$ | Oceanic | 8.1 |
| $NH_3$ | Anthropogenic | 41.6 |
| $NH_3$ | Biomass | 3.9 |
| $NO_x$ | Soil | 5.5 |
| $NO_x$ | Anthropogenic | 35.8 |
| $NO_x$ | Biomass | 7.5 |
| $NO_x$ | Aircraft | 0.7 |

## 4 Results and discussion

In the following section, we present the impact of LNO$_x$ on tropospheric composition and radiation with and without nitrate aerosol.

### 4.1 Modelled lightning flash rate and LNO$_x$

Luhar et al. (2021) previously evaluated the flash-rate parameterisations Eq. (2) – (6) within their ACCESS-UKCA model (which is UM-UKCA vn8.4 (with GA4.0) and which does not have nitrate aerosol) using the Lightning Imaging Sensor (LIS) / Optical Transient Detector (OTD) satellite data of global flash-rate distribution (Cecil et al., 2014). Here, we have used the above flash-rate parameterisations in UM-UKCA at vn13.2 (with GA8.0), both with and without nitrate – this model version has essentially the same convection scheme as in vn8.4 and therefore results in similar global flash-rate

distributions. Here, we only do a limited comparison of the simulated flash-rate density with the LIS/OTD climatology since a comprehensive evaluation (including a seasonal comparison) has already been done by Luhar et al. (2021).

The LIS/OTD flash density climatology in Figure 1a shows high overland values in the tropics and subtropics, as well as in lower midlatitudes. Some relatively high levels are also observed over water at these latitudes, especially over the western Atlantic, Pacific, western Indian Ocean close to southern Africa, and the waters around the maritime continent. The

modelled flash density distribution in Figure 1b obtained using the PR92 scheme is able to reproduce the broad observed distribution at low latitudes over land (barring parts of India), however, it is apparent that it does not simulate well the extension of the observed flash density over the temperate latitudes, especially in the Northern Hemisphere (NH). It is also clear that the PR92 scheme predicts nearly zero marine flash density, contrary to the observations. On the other hand, the Lu21 scheme reproduces the observed distribution of lightning flash density over the ocean considerably better compared to

the PR92 scheme (Figure 1b), although some significant spatial differences are visible (for example, high bias over the Pacific and equatorial Indian Ocean, and the low bias over western Indian Ocean close to southern Africa) compared to the LIS/OTD climatology. The simulated overland distributions in Figure 1b and Figure 1c are very similar.

The simulated flash-rate distributions in Figure 1b and Figure 1c show a few small areas with relatively high flash-rate in the western equatorial Pacific Ocean (i.e., mostly areas east of the Philippines), and this is most likely because the land-sea mask

used at N96 resolution in the present model to distinguish between land and ocean for flash-rate calculation treat these grid point areas as land. Noting that there do not seem to be corresponding high flash-rate spots in the observations (Figure 1a), we recommend in future versions of UM-UKCA that these grid points be treated as water in the land-sea mask (or alternatively a minimum land fraction threshold for lightning onset could be applied).

With the present UM-UKCA, the total global lightning flash frequency obtained from the PR92 scheme without nitrate is

27.7 flashes s$^{-1}$, of which 27.2 flashes s$^{-1}$ is over the land and 0.5 flashes s$^{-1}$ is over the ocean. The corresponding values for the Lu21 scheme are 42.5, 30.9 and 11.6 flashes s$^{-1}$. For comparison, the values obtained from the LIS/OTD climatology are

46.3, 38.5 and 7.7 flashes s⁻¹ (Cecil et al., 2014; Luhar et al., 2021), respectively, which show a considerable improvement in the flash-rate estimate by the Lu21 scheme over the ocean. In the study by Luhar et al. (2021) using UM at vn8.4, the same PR92 scheme yielded 32.9, 32.5 and 0.4 flashes s⁻¹, respectively, and the same Lu21 scheme yielded 45.0, 35.9 and 9.1 flashes s⁻¹, respectively. These differences in the flash rate between the two model versions show the impact of incremental changes made in the model, although the approach used in the convection scheme remains the same in both model versions.

When nitrate is explicitly added to the model, the PR92 scheme gives 27.3 flashes s⁻¹ and the Lu21 scheme gives 42.0 flashes s⁻¹ for the globe. Therefore, there is a very small decrease (~ 1.3%) in flesh frequency for both PR92 and Lu21 schemes with nitrate in UM-UKCA, which demonstrates how inclusion of nitrate aerosol may influence lightning flash rate itself. While this is interesting, we have not investigated this further given that these are atmospheric-only simulations and ocean feedbacks may change the impact of nitrate on flash rate. Note that this nitrate-lightning feedback may be due to indirect effects of nitrate aerosol through cloud microphysics and radiation that impact model's meteorology and convection scheme (note also that there is a high sensitivity of the parameterised flash rate to cloud-top height due to its almost 5th power dependence on the latter, so even minute variations in cloud-top height can cause significant changes in the calculated flash rate). In this context, there is some work done by Wang et al. (2018) on the effects of aerosols on NOₓ production by lightning via direct and indirect aerosol effects.

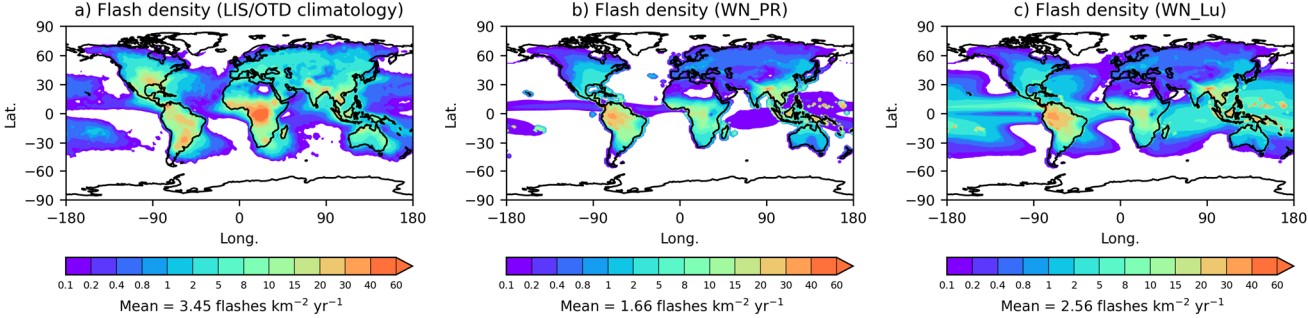

**Figure 1: Global distribution of the mean lightning flash density (flashes km⁻² yr⁻¹): (a) the LIS/OTD satellite climatology, (b) the modelled distribution using the PR92 flash-rate parameterisations, and (c) modelled distribution using the Lu21 flash-rate parameterisations (both with nitrate aerosol included).**

Since $P_{NO}$ has a fixed value, the above differences in the modelled flash rate are reflected in nearly linear proportion in the global LNOₓ amounts obtained from the various model runs (see Table 2).

## 4.2 Impact of nitrate aerosol on gas-phase tropospheric composition

The importance of LNO$_x$ on gas-phase tropospheric composition has previously been demonstrated by several studies (e.g., Labrador et al., 2005, Gordillo-Vázquez et al., 2019, and Luhar et al., 2021). Below, we examine the impact of inclusion of nitrate on the modelled gas-phase tropospheric composition, viz. total O$_3$, NO$_x$, OH and CO and methane lifetime, by considering the 'no-nitrate' (NN_Lu) and 'with-nitrate' (WN_Lu) runs involving the Lu21 flash-rate parameterisation. (Table 4 summarises the global averages of various atmospheric fields obtained from the no-nitrate and with-nitrate simulations for the three lightning NO$_x$ options. The change per Tg[N] yr$^{-1}$ of LNO$_x$ based on the slopes of linear least-squares fits is also given).

The modelled near-surface (at the lowest 20-m model level) O$_3$ mixing ratio (ppbv) distribution in Figure 2a and b suggests that both NN_Lu and WN_Lu cases look qualitatively very similar, showing relatively high levels in the Northern Hemisphere, particularly within 0–50°N which can be associated with the higher precursor emissions in these regions. However, it is apparent from the difference plot (Figure 2c) that there is a global decrease of near-surface O$_3$ when nitrate is included. This decrease is as much as 3 ppbv within the above latitudinal region, and on-average it is 4.3% (from the global mean of 29.9 ppbv). Annually averaged data from our 15-year model simulation (sample size = 15) were used in a t-test to determine whether the means of two populations differed significantly at a 95% confidence level. Figure 2c also displays hatched areas of statistically insignificant difference (so significance level less than 5%) which suggest that the differences are significant over most of the globe.

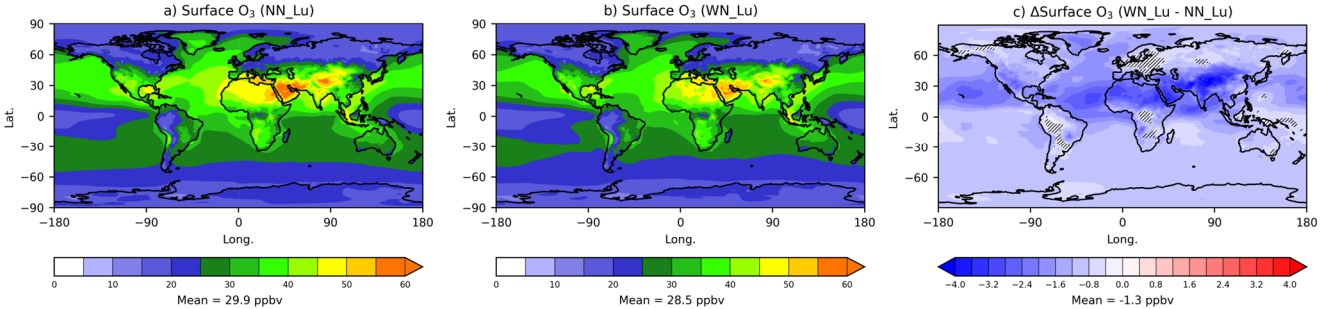

**Figure 2: Near-surface ozone concentration (ppbv) modelled (a) without (NN_Lu) and (b) with (WN_Lu) the nitrate scheme. The difference (WN_Lu - NN_Lu) is shown in (c). The Lu21 lightning flash-rate parameterisation was used. The difference plot (c) also shows hatched areas representing statistically insignificant difference (significance level less than 5%).**

**Table 4: Modelled global averages of various atmospheric variables obtained from the no-nitrate (NN) and with-nitrate (WN) simulations for three lightning $NO_x$ setup options: no $LNO_x$, Price and Rind's (1992) (PR92 or PR) lightning scheme and Luhar et al. (2021) (Lu21 or Lu) lightning scheme. All species are tropospheric averages, aerosol no. is aerosol number concentration, nu = nucleation mode, Ai = Aitken mode, Ai (in) = Aitken insoluble mode, ac = accumulation mode, co = coarse mode.**

| Simulation | Global variable | Lightning scheme | | | Change per |
|---|---|---|---|---|---|
| | | None | PR92 | Lu21 | Tg[N] yr$^{-1}$ of $LNO_x$ |
| No nitrate | $LNO_x$ emission (Tg[N] yr$^{-1}$) | 0 | 3.41 | 5.24 | - |
| | $O_3$ burden (Tg) | 278.5 | 325.2 | 348.8 | 13.45 |
| | $O_3$ (ppbv) | 48.1 | 56.7 | 60.9 | 2.46 |
| | OH ($\times 10^5$ molec. cm$^{-3}$) | 8.95 | 11.32 | 12.63 | 0.70 |
| | Methane lifetime (yr) | 8.55 | 7.49 | 7.03 | -0.35 |
| | CO (ppbv) | 93.7 | 82.4 | 78.1 | -3.03 |
| | NO (pptv) | 14.2 | 20.1 | 23.9 | 1.85 |
| | $NO_2$ (pptv) | 36.4 | 45.7 | 51.6 | 2.89 |
| | $NH_3$ (pptv) | 177.0 | 176.0 | 176.5 | -0.12 |
| | $HNO_3$ (pptv) | 151.0 | 190.2 | 213.8 | 11.9 |
| | $N_2O_5$ (pptv) | 0.41 | 0.63 | 0.78 | 0.070 |
| | $NO_3$ radical (pptv) | 0.37 | 0.45 | 0.50 | 0.023 |
| | $SO_4$ burden (µg[S] m$^{-2}$) | 981.5 | 1019.8 | 1036.2 | 10.54 |
| | Aerosol no.: nu (cm$^{-3}$) | 1749.8 | 1821.4 | 1890.8 | 26.16 |
| | Aerosol no.: Ai (cm$^{-3}$) | 331.2 | 360.0 | 377.8 | 8.84 |
| | Aerosol no.: Ai (in) (cm$^{-3}$) | 8.64 | 8.70 | 8.67 | 0.0065 |
| | Aerosol no.: ac (cm$^{-3}$) | 39.1 | 40.7 | 41.4 | 0.510 |
| | Aerosol no.: co (cm$^{-3}$) | 0.154 | 0.155 | 0.155 | 0.0002 |
| | $AOD_{550}$ | 0.1393 | 0.1407 | 0.1410 | 0.00034 |
| | $R_n^{TOA}$ (W m$^{-2}$) | 0.50 | 0.64 | 0.71 | 0.040 |
| | CDNC (cm$^{-3}$) | 6.38 | 6.49 | 6.53 | 0.029 |
| With nitrate | $LNO_x$ emission (Tg[N] yr$^{-1}$) | 0 | 3.36 | 5.18 | - |
| | $O_3$ burden (Tg) | 260.8 | 307.1 | 332.6 | 13.85 |
| | $O_3$ (ppbv) | 44.9 | 53.5 | 58.1 | 2.55 |
| | OH ($\times 10^5$ molec. cm$^{-3}$) | 8.20 | 10.61 | 12.01 | 0.73 |
| | Methane lifetime (yr) | 9.15 | 7.90 | 7.38 | -0.29 |
| | CO (ppbv) | 98.5 | 85.4 | 80.4 | -3.55 |

| | | | | |
|---|---|---|---|---|
| NO (pptv) | 12.7 | 18.1 | 22.1 | 1.79 |
| $NO_2$ (pptv) | 33.5 | 42.1 | 48.2 | 2.80 |
| $NH_3$ (pptv) | 29.8 | 27.9 | 27.2 | -0.51 |
| $HNO_3$ (pptv) | 87.9 | 122.2 | 144.0 | 10.8 |
| $N_2O_5$ (pptv) | 0.31 | 0.51 | 0.65 | 0.065 |
| $NO_3$ radical (pptv) | 0.32 | 0.39 | 0.44 | 0.023 |
| $SO_4$ burden ($\mu g[S]\ m^{-2}$) | 995.1 | 1035.5 | 1052.7 | 11.23 |
| $NH_4$ burden ($\mu g[N]\ m^{-2}$) | 814.3 | 831.3 | 837.0 | 4.47 |
| Fine $NO_3$ burden ($\mu g[N]\ m^{-2}$) | 291.3 | 301.9 | 304.8 | 2.68 |
| Coarse $NO_3$ burden ($\mu g[N]\ m^{-2}$) | 124.7 | 134.5 | 139.7 | 2.90 |
| Aerosol no.: nu ($cm^{-3}$) | 1721.1 | 1807.5 | 1875.1 | 29.23 |
| Aerosol no.: Ai ($cm^{-3}$) | 302.0 | 333.9 | 353.6 | 9.90 |
| Aerosol no.: Ai (in) ($cm^{-3}$) | 9.10 | 9.08 | 9.13 | 0.0043 |
| Aerosol no.: ac ($cm^{-3}$) | 42.9 | 44.8 | 45.5 | 0.443 |
| Aerosol no.: co ($cm^{-3}$) | 0.163 | 0.166 | 0.166 | 0.00062 |
| $AOD_{550}$ | 0.1539 | 0.1546 | 0.1553 | 0.00026 |
| $R_n^{TOA}$ ($W\ m^{-2}$) | 0.11 | 0.25 | 0.29 | 0.036 |
| CDNC ($cm^{-3}$) | 6.59 | 6.73 | 6.80 | 0.041 |

In the model, the tropopause is defined as follows. In the extratropics (latitude ≥ |28|°), the tropopause is the pressure level of the 2-PVU (potential vorticity units) surface, and in the tropics (latitude ≤ |13|°) it is the pressure level of the 380 K potential temperature isentropic surface. Between the two latitudes, a weighted average of the two definitions is used following the method of Hoerling et al. (1993). The zonal mean tropospheric $O_3$ from the two runs also look very similar qualitatively (Figure 3a and b) with less $O_3$ in the lower troposphere (altitudes < 5 km) over the Southern Hemisphere (SH) compared to

that over the Northern Hemisphere. Inclusion of nitrate results in $O_3$ decreases throughout the troposphere (Figure 3c) with the biggest decreases being located in the mid to upper troposphere in the Northern Hemisphere. The volume-weighted global mean tropospheric $O_3$ obtained with nitrate is 58.1 ppbv which is 4.6% lower than with no nitrate. Similarly, the tropospheric ozone burden decreases by the same relative amount (by 16 Tg from 348.8 Tg to 332.8 Tg) with nitrate (see Table 2). Recently, an evaluation study by Russo et al. (2023) involving UM-UKCA (without nitrate) and focusing on the

North Atlantic region suggests a considerable model positive bias for ozone in the tropical upper troposphere. They attribute this to shortcomings of the model's convection and lightning parameterisations (the latter based on the convective cloud-top height approach), which underestimate lightning flashes in mid-latitudes relative to the tropics, which was also evident in the study by Luhar et al. (2021). However, it is clear from Figure 3c that the inclusion of nitrate in the model may alleviate model's positive ozone bias in the tropical upper troposphere to some extent.


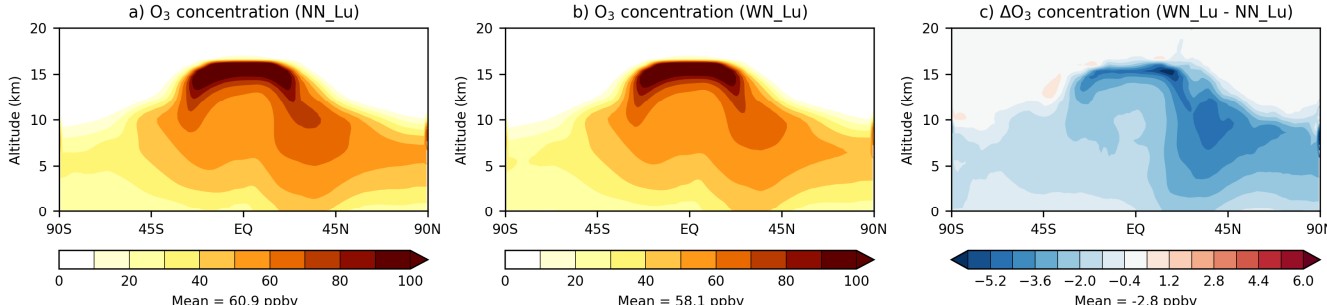

**Figure 3: Zonal mean tropospheric ozone (ppbv) modelled (a) without and (b) with the nitrate scheme. The difference between (b) and (a) is shown in (c). The Lu21 lightning flash-rate parameterisation was used.**

OH is the controlling oxidant in the global atmosphere and governs the chemical lifetime of most anthropogenic and natural gases, for example $CH_4$ and CO. Concentration of OH in the troposphere is governed by a complex set of chemical reactions involving species such as tropospheric $O_3$, $CH_4$, CO, nonmethane VOCs and $NO_x$, as well as the amount of humidity and solar radiation (e.g., Naik et al., 2013). The nitrate aerosol mechanism is linked to some of these reactions (e.g., those involving $HNO_3$) which leads to changes tropospheric OH. The simulated zonal average tropospheric OH in Figure 4a shows

high OH levels near the in the tropical lower troposphere, particularly near the surface within 10–30° N with values as high as $(20–25) \times 10^5$ molecules $cm^{-3}$. The OH concentrations decline with height, but a secondary maximum in the tropical upper

troposphere (at ~ 13 km) is apparent, likely linked to $LNO_x$ emissions. The corresponding Figure 4b with nitrate indicates some decrease in OH, and this is more apparent in the difference plot in Figure 4c where there is a decrease in OH everywhere with the largest changes being as much as $1.3 \times 10^5$ molecules cm$^{-3}$ within 0–40° N in the Northern

Hemisphere. Overall, there is a decline of 5.0% in the global tropospheric OH when nitrate is included in the model.

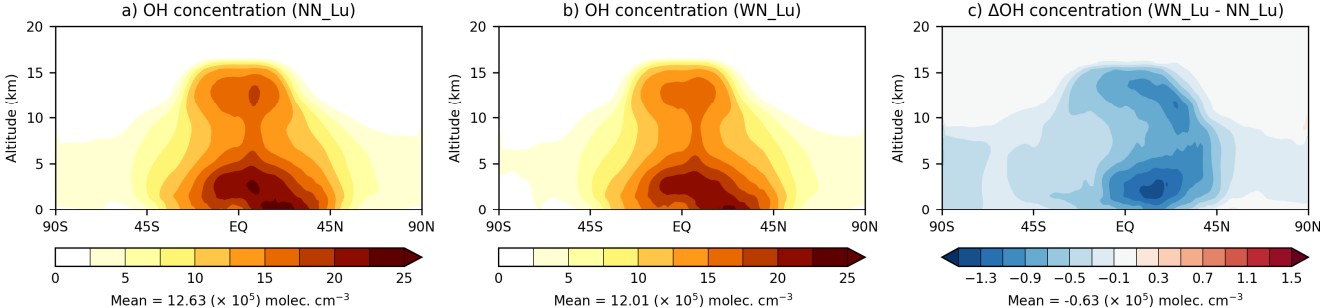

**Figure 4: Zonal mean tropospheric OH ($\times 10^5$ molecules cm$^{-3}$) simulated (a) without and (b) with the nitrate scheme. The difference between (b) and (a) is shown in (c). The Lu21 lightning flash-rate parameterisation was used.**


As a consequence of the decrease in OH with nitrate in the model, the methane lifetime ($\tau_{CH_4}$) with respect to loss by OH in the troposphere increases by 5% from 7.0 years, and on average the tropospheric CO goes up by 3.0% from the global average value of 78.1 ppbv, as compared to the simulation with no nitrate. On the other hand, there is a decrease in NO by 7.5% and that in $NO_2$ by 6.7%, owing to the swift removal of particulate nitrate from the atmosphere compared to gas phase

nitrogen species. In Figure 5a, $\tau_{CH_4}$ plotted as a function of $LNO_x$ for all simulations (Table 2) suggests that within the range of $LNO_x$ considered, on average, $\tau_{CH_4}$ is greater by ~ 0.4 years when nitrate is included. This approximately equates to the increase in $\tau_{CH_4}$ as a result of per unit Tg[N] yr$^{-1}$ decrease in $LNO_x$ (using the slopes of the linear best fit lines in Figure 5a). The 1-sigma standard deviation uncertainty in $\tau_{CH_4}$ in Figure 5a (and in Table 2) calculated from the annual means over 15 years of simulation after linear trending is on average ± 0.027 years which is much smaller than the mean decrease of ~ 1.7

years within the range of $LNO_x$ considered. In Figure 5b, the reduction in the ozone burden in the troposphere when using the nitrate scheme is almost uniform at 16.5 Tg $O_3$ across the $LNO_x$ values considered, and this magnitude is comparable to about 13 Tg $O_3$ generated per unit Tg[N] yr$^{-1}$ of $LNO_x$. This result is perhaps somewhat contrary to that reported by Tost (2017) who deduced that the nitrate formation does not have important influence on the distribution of tropospheric ozone. The tropospheric $O_3$ burden obtained by Luhar et al. (2020, 2021) with an older version of UM-UKCA (without nitrate) that

used the Lu21 scheme (giving an $LNO_x$ of 6.6 – 6.9 Tg[N] yr$^{-1}$) was 304 – 308 Tg $O_3$, which is considerably lower than 348.8 Tg $O_3$ obtained from the corresponding present simulation NN_Lu. Notwithstanding the various differences between the two simulations (e.g. differences in prescribed emissions, updated model schemes and parameters, nudging etc.), one

important reason for this difference is that the old version of UKCA StratTrop included the chemical reaction $HO_2 + NO \rightarrow$ $HNO_3$ which acts as a significant sink of $HO_x$ (OH, $HO_2$) in the upper troposphere, and thus reducing ozone. However, based

on further studies (e.g., Mertens et al., 2022), it was determined that this reaction was not important and was omitted in later versions of StratTrop. The uncertainty in the ozone burden in Figure 5b (and in Table 2) is on average ± 2.9 Tg $O_3$, much smaller than the mean increase of ~ 70 Tg $O_3$ within the $LNO_x$ range considered.

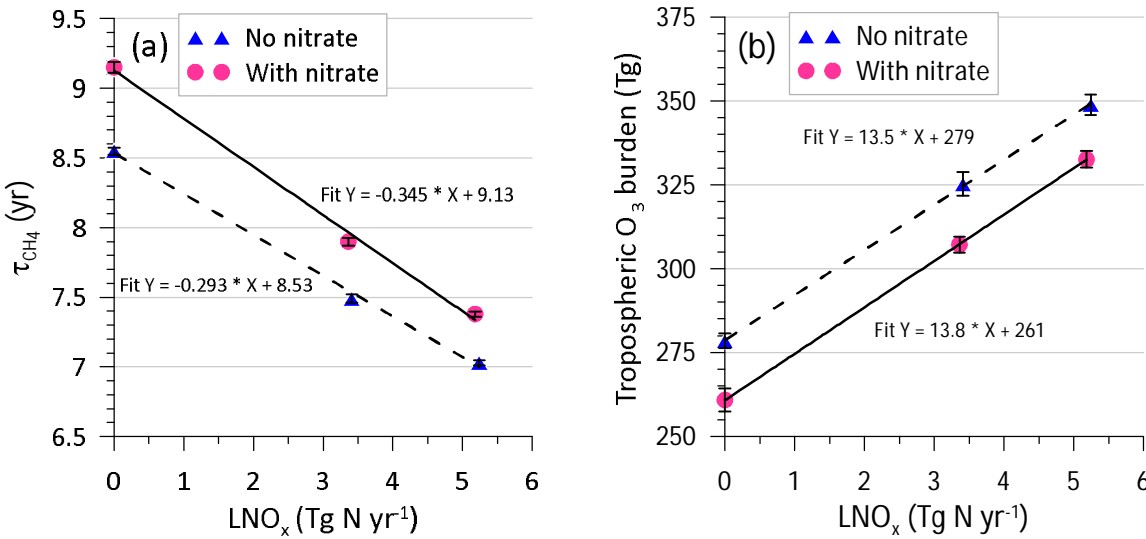

**Figure 5: (a) Modelled methane lifetime ($\tau_{CH_4}$) due to loss by tropospheric OH, as a function of lightning-generated $NO_x$, and (b) the same plot but for tropospheric $O_3$ burden. The lines are linear least squares fits and the error bars correspond to a 1-sigma standard deviation.**

When nitrate aerosol is considered, there is a decrease in $HNO_3$ mixing ratio throughout the troposphere (Figure 6), mostly in

the lower troposphere and between 0–45° N (where most of the surface $NH_3$ emissions are located), as part of the gas-phase $HNO_3$ is used up in the nitrate aerosol mechanism (following neutralisation by the uptake of ammonia). On average, this reduction is 33% compared to the no-nitrate value. The distribution of the reduction in Figure 6c is qualitatively very similar to that in $NH_3$, with the latter decreasing by 85% (plot not shown) compared to the no-nitrate case. We find that on average 1 Tg[N] yr$^{-1}$ of $LNO_x$ emission results in a 11 pptv increase in tropospheric nitric acid mixing ratio with or without nitrate.


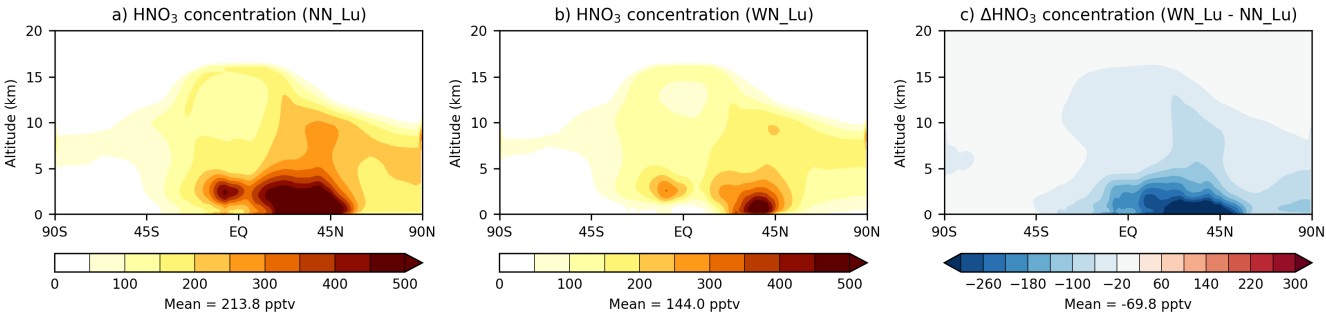

**Figure 6: Zonal mean tropospheric HNO₃ (pptv) modelled (a) without and (b) with the nitrate scheme. The difference between (b) and (a) is shown in (c). The Lu21 lightning flash-rate parameterisation was used.**

We have also looked at the gaseous nitrate radical (NO₃) (different from nitrate aerosol) and N₂O₅ (dinitrogen pentoxide) mixing ratios. The NO₃ radical has relatively low concentration but it is the controlling nighttime oxidant (the counterpart of OH in daytime) (Archer-Nicholls et al., 2023). It is generated by the reaction involving NO₂ and O₃ and reacts rapidly with unsaturated VOCs (e.g. isoprene and terpenes), thus affecting their atmospheric budgets as well as their degradation products (Khan et al., 2015). The NO₃ radical reacts further with NO₂ during nighttime to form N₂O₅ (dinitrogen pentoxide), another

important nighttime species whose uptake onto aerosol removes NOₓ from the atmosphere and produces nitric acid (Brown et al., 2006). We find that with the nitrate aerosol scheme, NO₃ radical is decreased by 11.5% overall from the tropospheric mean value of 0.5 pptv and, similarly, N₂O₅ goes down by 16.4% from the tropospheric mean value of 0.78 pptv.

### 4.3 Impact of LNOₓ on aerosol mass concentration

Previous modelling studies have shown that lightning NOₓ does lead to a significant increase in particulate nitrate

concentrations (Tost, 2017). Jones et al. (2021) have presented results with nitrate aerosol that were obtained with the PR92 scheme, but without explicitly looking at sensitivity to lightning. Tost (2017) has investigated no-LNOₓ and LNOₓ (with the PR92 scheme) cases with nitrate included. We now investigate whether the changing of the LNOₓ parameterisation (to Lu21) significantly impacts NO₃ concentrations. The three simulations (no-LNOₓ, PR92 and Lu91) with and without nitrate also provide three amounts of global LNOₓ against which the globally averaged parameter values can be plotted.

Ammonium nitrate aerosol is formed from HNO₃ condensation onto pre-existing aerosols, mainly thermodynamically stabilised by ammonium (NH₄⁺). The modelled column annual-mean mass burdens of NH₄ (from both (NH₄)₂SO₄ and NH₄NO₃), fine NO₃ (from NH₄NO₃), coarse NO₃ (from both NaNO₃ and Ca(NO₃)₂) and SO₄ from the PR92 and Lu21 simulations and the differences between the two simulations (Lu21 – PR92) are shown in Figure 7 (top two rows). As fine NO₃ is associated with NH₄, their spatial distributions are nearly the same, with relatively high burdens over land in the

Northern Hemisphere, particularly the values as high as 3 mg[N] m$^{-2}$ in South Asia, 2 mg[N] m$^{-2}$ in East Asia/China, and 1.1 mg[N] m$^{-2}$ in central North America.

Moving from the PR92 to Lu21 scheme, there is a very small global mean increase (~ 1%) in both $NH_4$ and fine $NO_3$ burdens as $LNO_x$ increases from 3.36 to 5.18 Tg[N] yr$^{-1}$. This increase is dominated by particle mass in the Aitken mode for $NH_4$ (68%) and accumulation mode for fine $NO_3$ (77%). There is a greater increase in the coarse $NO_3$ (as $NaNO_3$), at ~ 4%.

However, the global difference plots in Figure 7 show considerable regional variations in the changes in aerosol burden, with the Lu21 scheme predicting larger fine nitrate concentrations over south-eastern China (by 90 µg[N] m$^{-2}$), North America and Central Africa (by 40 µg[N] m$^{-2}$), and western Europe (by 50 µg[N] m$^{-2}$). In contrast, the Lu21-predicted nitrate is lower over India (by as much as 100 µg[N] m$^{-2}$), north-east Asia and Eastern Europe. Very similar qualitative difference distribution is obtained for $NH_4$. For coarse nitrate, there are increases over most of the globe, with larger magnitudes in the tropics (by ~15 µg[N] m$^{-2}$) compared to the PR92 scheme.

Sulfate aerosol is produced when OH oxidises $SO_2$ in the presence of water vapour to form $H_2SO_4$ which either nucleates or condenses on existing particles, depending upon its concentration. The distributions of $SO_4$ burden in Figure 7 (bottom panels) show a global increase of ~ 1.7% with the Lu21 scheme and this increase is dominated by the particles in the Aitken mode (55%) and accumulation mode (42%). There are relatively large increases (~ 70 µg[S] m$^{-2}$) over most of the tropics, but burdens decrease over Russia, India, China, and north-east Asia.

The tropospheric burdens of $NH_4$, fine and coarse $NO_3$, and $SO_4$ for the Lu21 scheme with nitrate are 0.43 Tg[N], 0.16 Tg[N], 0.07 Tg[N] and 0.54 Tg[S], respectively. These values can be compared with the global $NH_4$, fine-mode $NO_3$ and $SO_4$ burdens of 0.13–0.58 Tg[N], 0.03–0.42 Tg[N], and 0.28–1.10 Tg[S], respectively, obtained from the AeroCom phase III model intercomparison study by Bian et al. (2017).

The above differences can be compared with those between the Lu21 scheme and when $LNO_x$ is set to zero (which means the global $LNO_x$ difference between the two cases is 5.18 Tg[N] yr$^{-1}$). Figure 8 is the same as Figure 7(right), except that the difference is between the Lu21 scheme and the no-$LNO_x$ case. Hached areas representing statistically insignificant difference (significance level less than 5%) are also plotted. On global average, there is a mean increase of 2.8% in $NH_4$, 4.7% in fine $NO_3$, 12% in coarse $NO_3$, and 5.8% in $SO_4$. This increase is dominated by particle mass in the Aitken mode for $NH_4$ (68%), accumulation mode for fine $NO_3$ (77%), coarse mode of $NaNO_3$ (73%) and Aitken mode for $SO_4$ (66%). Taken together Figure 7 and Figure 8 indicate that updating the $LNO_x$ parameterisation from PR92 to Lu21 has a substantial impact on the $LNO_x$-mediated nitrate aerosol concentrations.

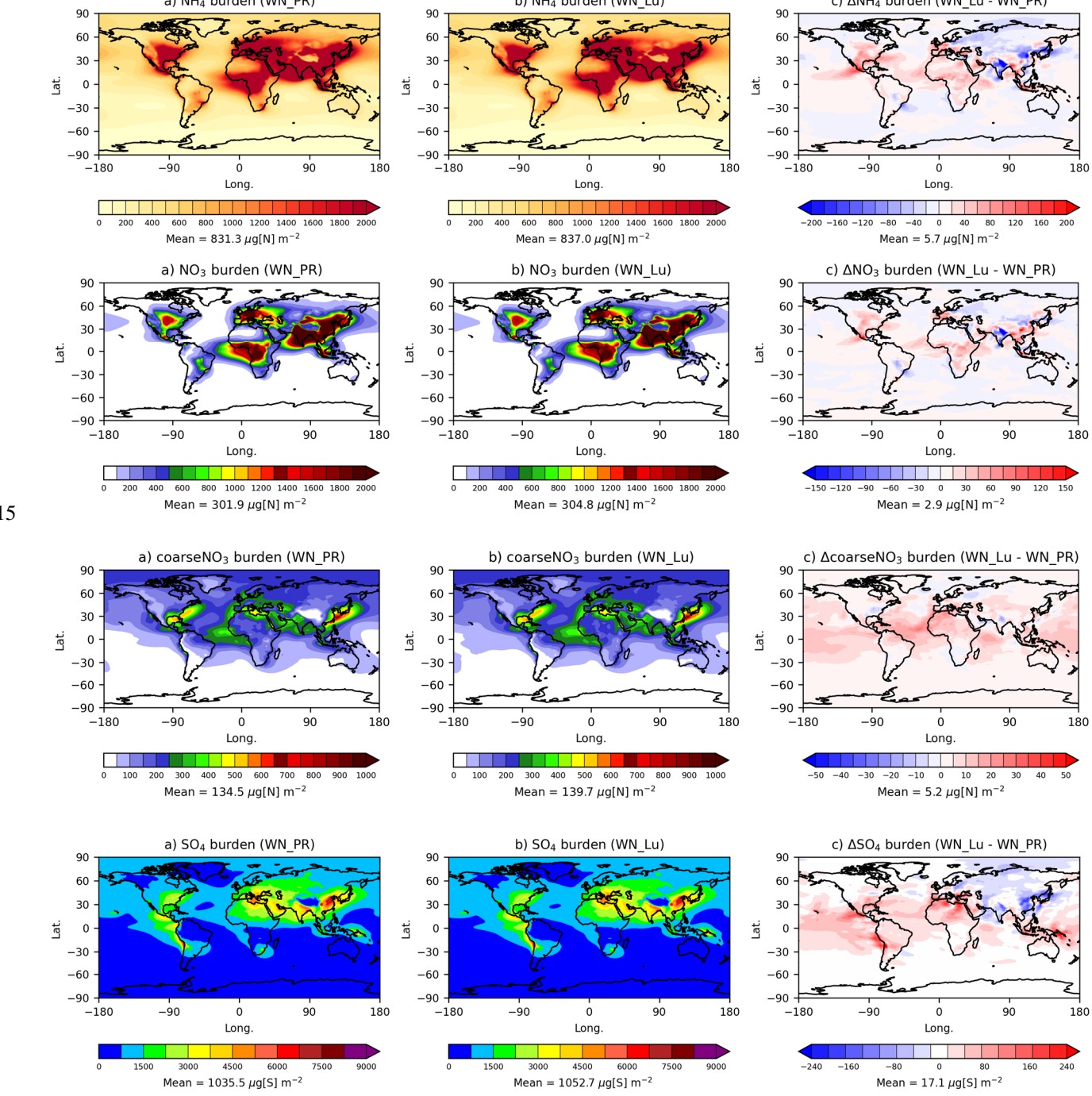

**Figure 7: Annual-mean tropospheric mass burdens of NH₄, fine NO₃, coarse NO₃, and SO₄ from the PR92 (left) and Lu21 (middle) simulations (both with nitrate) and the difference (right) between the two simulations.**



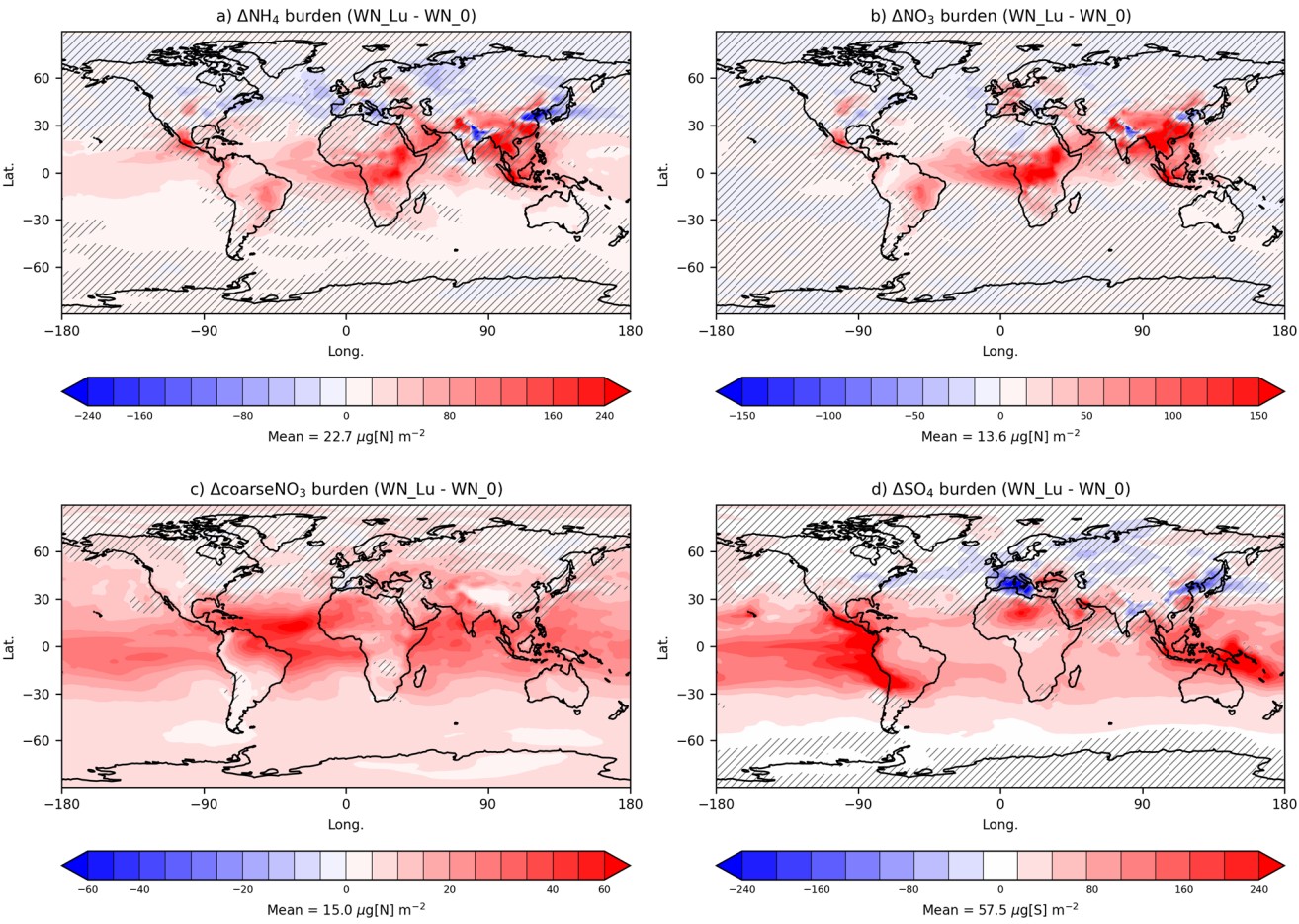

**Figure 8: Difference between the Lu21 and no-LNO$_x$ simulations (both with nitrate) for annual-mean tropospheric (a) NH$_4$, (b) fine NO$_3$, (c) coarse NO$_3$, and (d) SO$_4$ burdens. The hatched areas represent statistically insignificant difference (significance level less than 5%).**


The zonal-mean vertical distributions of NH$_4$, fine NO$_3$, coarse NO$_3$ (related to sea salt and dust) and SO$_4$ mass concentrations from the Lu21 simulation (with nitrate) shown in Figure 9 (left) suggest that these species are mostly confined to the lower troposphere (as their precursors are emitted from sources close to the surface). High concentrations are predicted near the surface between the equator and 60° N which correspond to the locations of the main precursor emissions

(i.e., NH$_3$ and NO$_x$). It is apparent that the vertical extent of fine NO$_3$ is limited because of the rapid wet removal of NH$_3$ gas from the atmosphere due to its high effective solubility, which in turn limits the vertical range to which NH$_4$NO$_3$ may establish by condensation. Conversely, near the surface over land regions NH$_4$NO$_3$ production is HNO$_3$ limited due to the much greater concentrations of NH$_3$ from agricultural sources (Jones et al., 2021). NH$_4$ would reach a greater altitude than

fine $NO_3$, due to its long-lived association with $SO_4$ aerosol. The coarse $NO_3$ ($NaNO_3 + Ca(NO_3)_2$) aerosol is also confined to

near the surface as they are easily removed from the atmosphere by wet deposition and gravitational settling.

The differences between the Lu21 and PR92 simulations (Lu21 – PR92) (Figure 9, middle) indicate that with the Lu21 scheme there are noticeable increases in $NH_4$ and fine $NO_3$ mass concentrations around the equator below 4 km and an elevated maximum at an altitude of 3 km at around 15° N. While most of the $LNO_x$ would be distributed vertically within the middle to upper troposphere in the tropics, $NH_3$ is mostly released at the surface between 0–60° N, and thus these are the

locations within the lower troposphere where $LNO_x$ results in an optimal formation of $NH_4$ and fine $NO_3$. This is clearer from the difference between the Lu21 and no-LNOx simulations (Lu21 – no-LNOx) (Figure 9, right) where the differences are much greater, but the qualitative shape of the difference patterns are very similar. This suggests that the differences in the distribution are governed by a balance between availability of $LNO_x$ and $NH_3$. There are areas of small decreases in the lower troposphere in the Northern Hemisphere beyond 40° N. With regards to the coarse $NO_3$, the increases are mostly

confined near the surface and symmetrically centred around the equator, which suggests that the differences in the distribution are dominated by $LNO_x$ availability. The bottom plots in Figure 9 generally show increases in $SO_4$ with the Lu21 scheme throughout the troposphere between 40° S – 40° N and areas of decrease below 3 km beyond 40° N. There is an amplification of this behaviour when looking at the differences between the Lu21 case with the no-LNOx case. The overall increase in $SO_4$ with the Lu21 scheme is possibly due to increase in tropospheric oxidants in response to increase in $LNO_x$

(for example, there is a 13% increase in OH with the Lu21 scheme compared to the PR92 scheme with nitrate included).

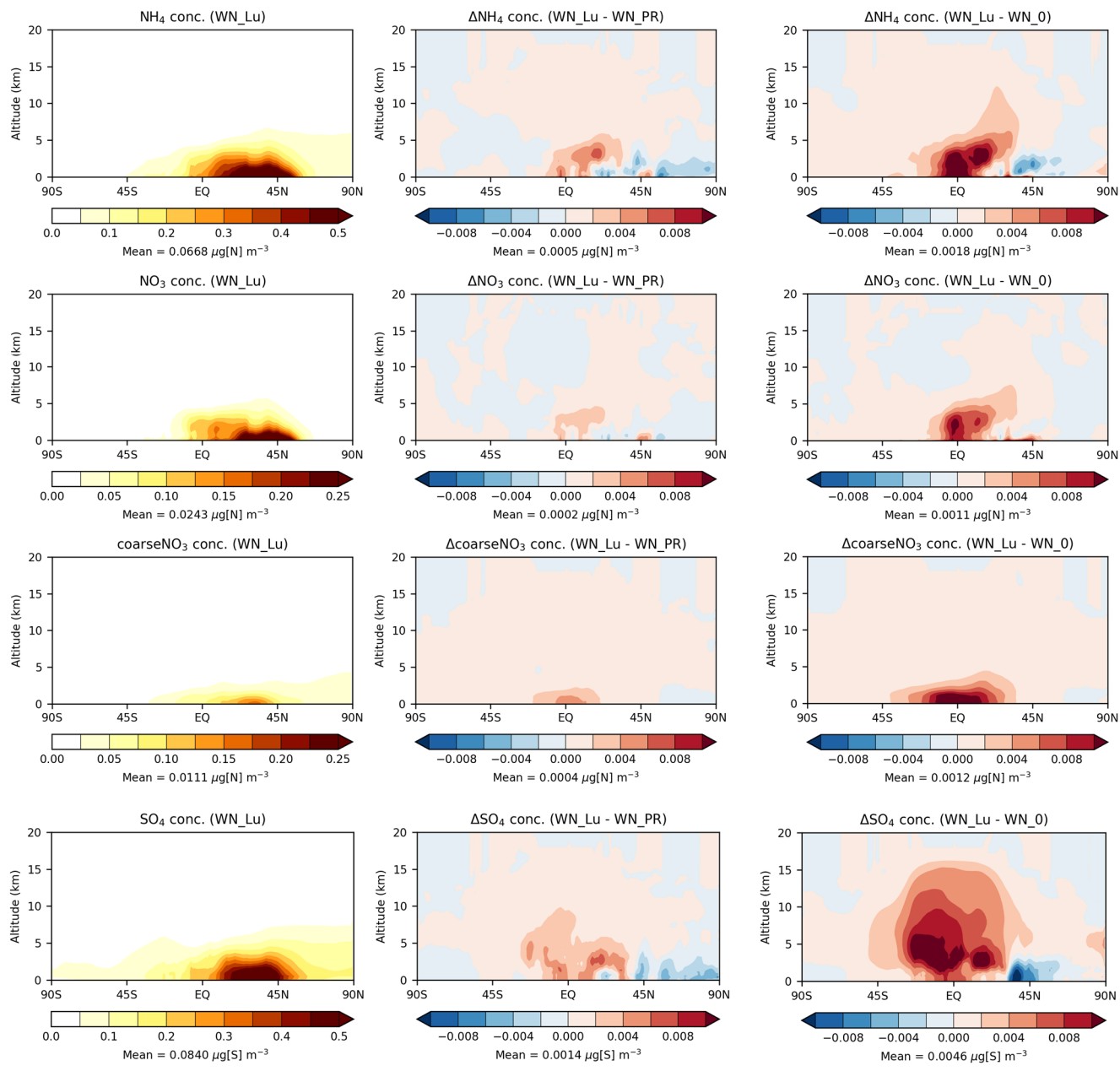

**Figure 9: Zonal mean tropospheric NH₄, fine NO₃, coarseNO₃, and SO₄ concentrations from the Lu21 simulation with nitrate (left), the difference between the Lu21 and PR92 simulations (Lu21 – PR92) (middle), and the difference between the Lu21 and no-LNOₓ simulations (Lu21 – no-LNOₓ) (right).**


## 4.4 Impact of LNO$_x$ on aerosol number concentration

In addition to the aerosol mass concentrations above, we can also examine zonal differences in prognostic aerosol number concentrations. Figure 10 presents the modelled zonal mean tropospheric aerosol number concentrations from the Lu21 simulation with nitrate (left), the difference between the Lu21 and PR92 simulations (Lu21 – PR92) (middle), and the difference between the Lu21 and no-LNOx simulations (Lu21 – no-LNOx) (right).

In Figure 10 (left), the nucleation mode (soluble) particles are the highest in number and are mostly located in the upper troposphere within 40° S – 40° N, followed by Aitken mode (soluble) particles with a maximum in the mid troposphere around the equator, and accumulation mode (soluble) particles in the lower troposphere between 30° S – 70° N. The coarse mode (soluble) particles (plot not shown) are the least in number and confined to very close to the surface due to their effective gravitational sedimentation, with more particles in the Southern Hemisphere than in the Northern Hemisphere (probably due to a larger oceanic surface in the SH so as to cause a large sea salt particle number concentration).

The difference plots in Figure 10 (middle) show that the Lu21 scheme with its higher amount of LNO$_x$ leads to a greater number concentration of nucleation mode particles (overall by about 3.6%) compared to the PR92 scheme, although there are regions, particularly near the tropopause between 30° S – 30° N, where the particle number concentration decreases with the Lu21 scheme. Both simulations include nitrate aerosol, but in the present scheme aerosol number concentrations are not modified explicitly by nitrate chemistry because of the assumption that NH$_4$ and NO$_3$ condense onto (and evaporates from) existing atmospheric aerosol rather than nucleating new particles. So, the changes in the number of nucleation mode particles in Figure 10 (middle) are likely due to an indirect impact of LNO$_x$ changes on the oxidising capacity of the atmosphere (for example, via changes in OH and O$_3$), increases in which would generally lead to enhanced formation of new particles owing to faster oxidation rates of gas-phase sulfur to sulfate conversion as LNO$_x$ is increased. (Enhancement of nucleation-mode particle numbers with lightning emissions in the upper troposphere was also predicted in the model simulations by Tost (2017) which included nitrate.)

In Figure 10 (middle), there is an overall increase of 5.6% in the number of particles in the Aitken mode in the troposphere with the Lu21 scheme and this increase is centred in the middle of the troposphere (~ 8 km) within 30° S – 30° N. The particle number in the accumulation mode Figure 10 (bottom) is confined to lower troposphere and is impacted much less (~ 1.5%) by the change in the LNO$_x$ scheme. There are negligible changes in the coarse mode and Aitken (insoluble) particle number concentrations with the changes in LNO$_x$ amounts considered (plots not shown).

The differences in the particle numbers between the Lu21 scheme and the no-LNO$_x$ case shown in Figure 10 (right) essentially show bigger increases but with qualitatively similar patterns – the increases in particle number in the nucleation, Aitken, and accumulation modes are 8.2%, 14.6% and 5.7%, respectively.

Thus, it is apparent from Figure 9 and Figure 10 that with the Lu21 scheme, with its 54% larger total $LNO_x$ compared to the PR92 scheme, there is a rise in the aerosol mass concentration (of $NH_4$, fine $NO_3$, coarse$NO_3$ and $SO_4$) in the lower troposphere whereas there is an increase in the total aerosol number concentration, particularly in the nucleation and Aitken modes, in the mid to upper troposphere. The magnitudes of these increases are amplified when comparing the results from the Lu21 scheme with the no-$LNO_x$ case.

Wang et al. (2021) observed formation of new ultrafine particles during lightning events, resulting in large increases in nucleation and Aitken mode aerosols (by 18.9 and 5.6 times, respectively) coupled with ~12% intensification of nitrate aerosol signals (but it is not clear from their data if there were any nitrate signals in the nucleation mode). These increases are qualitatively consistent with the modelled differences with and without lightning shown in Figure 10. In our model, nitrate does not participate in nucleation and the increases in particle numbers stem from $LNO_x$ enhancing the oxidizing

capacity of the troposphere and thus sulfate aerosol formation. Wang et al. (2021) point out that lightning channels may be conducive to an ion-induced nucleation process to generate new particles. Also, there is some evidence (e.g., Wang et al., 2020) that nitrate participates in nucleation directly. Note that any such direct new particle formation due to lightning or participation of nitrate in nucleation is not included in our model but may considered in future versions of the UM as new observations become available.


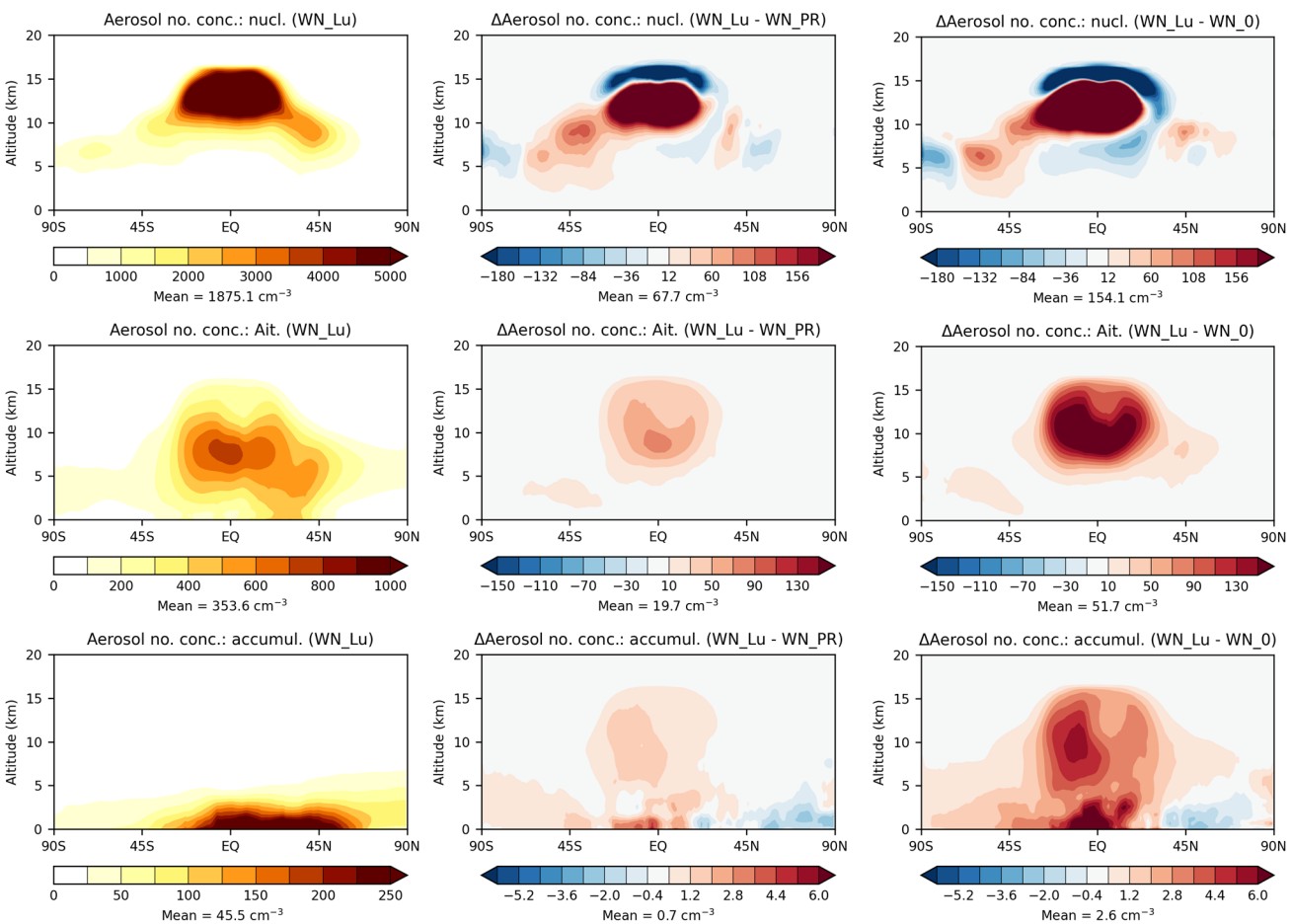

**Figure 10: Zonal mean tropospheric aerosol number concentrations from the Lu21 simulation with nitrate (left), the difference between the Lu21 and PR92 simulations (Lu21 – PR92) (middle), and the difference between the Lu21 and no-LNO$_x$ simulations (Lu21 – no-LNO$_x$) (right).**


It is also instructive to present particle number concentrations without and with nitrate. In Figure 10, we already presented the modelled zonal mean tropospheric aerosol number concentrations from the Lu21 simulation with the nitrate scheme. Figure 11 presents the difference between this simulation and the same simulation without the nitrate scheme. There are regions of both increased and decreased nucleation mode (soluble) particle number concentrations from mid to upper

troposphere with nitrate on, likely as a result of changes in oxidation rates of gas-phase sulfur to sulfate. But overall, there is a very small decrease of ~ 1% in nucleation model particle numbers with nitrate on. With nitrate on, there is a reduction in the Aitken mode particle number density (by 6.4% overall) throughout the troposphere, particularly between 10° – 50° N with higher concentrations in the lower troposphere. The probable reason for this is that condensational growth and altered

coagulation due the additional nitrate mass moves these particles to the accumulation mode. This is evident from the particle number distribution for the accumulation mode where there is a 10% enhancement in the particle number concentration with nitrate (mostly confined in the lower troposphere between 10° S – 50° N). Of all the modes, aerosols in the accumulation mode are of most importance as they have the greatest climate impact. They not only have the highest scattering efficiency, but they also have the longest atmospheric lifetime. They constitute the largest source of CCN.

The coarse mode aerosol number concentrations are much smaller (global tropospheric average = 0.16 cm$^{-3}$), are confined to near the surface, and the inclusion of nitrate leads to an increase of 7.1% over the no-nitrate value.

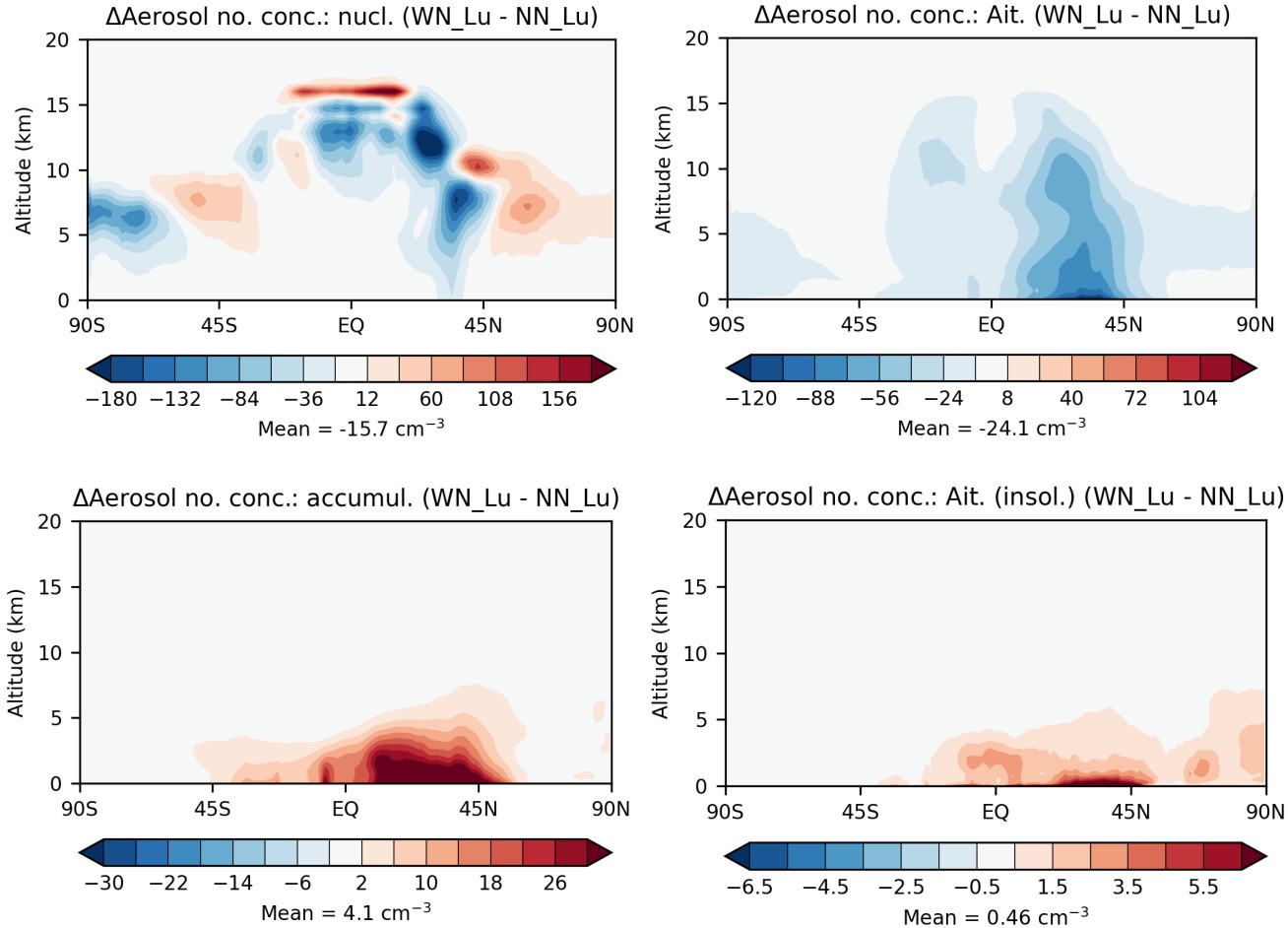

**Figure 11: Difference between the zonal mean tropospheric aerosol number concentrations in various aerosol modes from the Lu21 simulation with and without the nitrate scheme.**

We also looked at how tropospheric sulfate is impacted by the inclusion of nitrate, and this is presented in Supplement (S1).

## 4.5 Impact of LNOₓ on aerosol optical depth (AOD), TOA radiation, and CDNC

As demonstrated earlier, significant differences in the size distribution of aerosol in the troposphere are caused by lightning. In conjunction with the inclusion of nitrate, this has ramifications for the aerosol extinction and cloud properties (Tost, 2017). In Figure 12 (left), we present the modelled column aerosol optical depth ($AOD_{550}$ at 550 nm wavelength) from the Lu21 simulation with nitrate. Large AODs over central Africa (biomass burning region), north-west Africa (dust emission region), and China and India (anthropogenic pollution emissions) are evident. These patterns are consistent with the 2003–2012 mean MODIS (Collection 6) satellite $AOD_{550}$ data with a globally-averaged value of 0.162 (see Jones et al., 2021), which can be compared with the present model estimate of 0.155. The differences between the Lu21 and PR92 simulations (Lu21 – PR92) (Figure 12, middle) indicate that there is a very small (0.5%) overall increase in the global $AOD_{550}$ with the Lu21 scheme, but it is apparent there are larger differences regionally, for example an increase of 0.01–0.02 over north-western parts of Africa, western Europe, central Atlantic, and a decrease over India. The differences in $AOD_{550}$ between the Lu21 scheme and the no-LNOₓ case (both with nitrate) shown in Figure 12 (right) show, as expected, bigger increases (~ 1%).

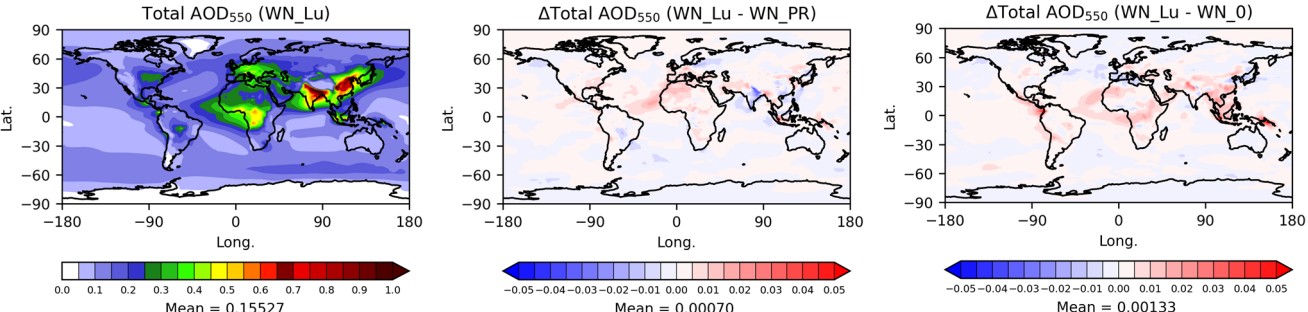

**Figure 12: Annual-mean aerosol optical depth ($AOD_{550}$) from the Lu21 simulation with nitrate (left), the difference between the Lu21 and PR92 simulations (Lu21 – PR92) (middle), and the difference between the Lu21 and no-LNOₓ simulations (Lu21 – no-LNOₓ) (right).**

In Figure 13, globally averaged $AOD_{550}$ plotted as a function of LNOₓ for all simulations suggests that within the range of LNOₓ considered, on average, $AOD_{550}$ is greater by ~ 0.015 (about 11%) when nitrate is included (this difference is almost thrice as large as that obtained by Jones et al. (2021)). Based on the slopes of the linear least squares fits, the increase in $AOD_{550}$ as a result of per unit Tg[N] yr⁻¹ increase in LNOₓ is $2.5 \times 10^{-4}$ with nitrate and $3.4 \times 10^{-4}$ without nitrate. The 1-sigma standard deviation uncertainty bars (calculated from the annual means over 15 years of simulation) in Figure 13 suggest that the mean increase in $AOD_{550}$ with LNOₓ within the range of LNOₓ considered is ~ 0.0015 which is comparable to or within the average $AOD_{550}$ uncertainty of ~ ± 0.0017.

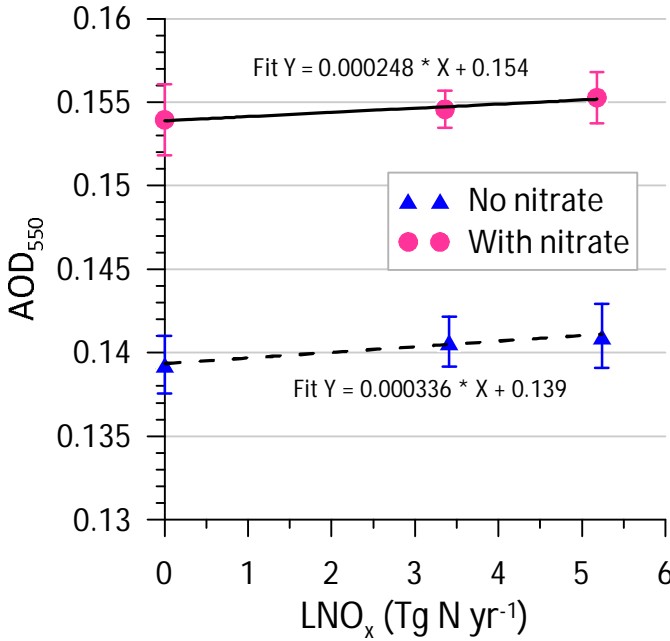

**Figure 13: Modelled globally averaged AOD$_{550}$ as a function of lightning-generated NO$_x$. The lines are linear least squares fits. The error bars correspond to a 1-sigma standard deviation.**

Lightning influences atmospheric radiation via changes in O$_3$ levels and methane lifetime, and through direct and indirect aerosol effects (e.g., Tost, 2017; Luhar et al., 2022). Figure 14 presents the modelled present-day all-sky annual-mean TOA net downward radiative flux ($R_n^{TOA}$), incorporating both shortwave and longwave components, as a function of LNO$_x$ for both with and without nitrate cases. In the present model simulation, perturbation in this quality is akin to the total effective radiative forcing (ERF) (Bellouin et al., 2020). Although these are based on only three model runs, the slopes of the linear best fit lines in Figure 14 indicates that there is a rise of 36 mW m$^{-2}$ in $R_n^{TOA}$ with a per Tg increase in N generated per year by lightning when nitrate is included in the model and it is 40 mW m$^{-2}$ without nitrate. (The latter is almost the same as 39.6 mW m$^{-2}$ (Tg[N] yr$^{-1}$)$^{-1}$ obtained by Luhar et al. (2022) using ACCESS-UKCA (an older version (v8.4) of UM-UKCA with somewhat different emission and model settings) without nitrate.) A decrease or negative change in $R_n^{TOA}$ signifies reduced atmospheric radiation absorption, indicating cooling conditions. In Figure 14, $R_n^{TOA}$ increases with LNO$_x$ which suggests that the positive radiative feedback from ozone increases outweighs the negative feedback from the reduction in methane lifetime and rise in aerosol concentrations as LNO$_x$ increases. This holds true whether nitrate is included or not. However, the inclusion of nitrate results in a change of approximately –0.4 W m$^{-2}$ in $R_n^{TOA}$ for any given LNO$_x$ level (Figure 14), indicating that the incorporation of nitrate in the model has a far greater impact on $R_n^{TOA}$ than the change in LNO$_x$ considered here. Note that the difference of –0.4 W m$^{-2}$ is nearly twice as large as that obtained by Jones et al. (2021) and this is likely due to the various updates to UM that have happened moving from science configurations GA7.1 and GL7.0 to GA8.0 and

GL9.0 (from UM vn11.8 to vn13.2), which include changes pertinent to aerosols, such as tuning of DMS emissions and
cloud droplet spectral dispersion parameterisation, and near-surface drag improvements (Jones et al., 2022). As a
comparison, in a regional model simulation study, Drugé et al. (2019) found that ammonium and nitrate aerosol caused a
TOA direct radiative forcing of about $-1.4$ W m$^{-2}$ under all sky conditions over Europe for the period 1979–2016. The
uncertainty bars in Figure 14 indicate that the increase in $R_n^{TOA}$ with LNO$_x$ within the range of LNO$_x$ considered is ∼ 0.20 W
m$^{-2}$ which can be compared within the average $R_n^{TOA}$ uncertainty of ∼ ± 0.125 W m$^{-2}$.


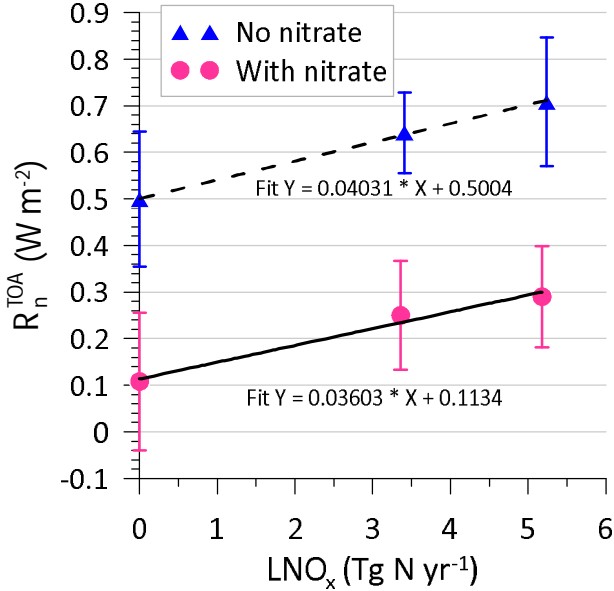

**Figure 14: Modelled annual-mean top-of-atmosphere (TOA) net downward radiative flux, as a function of lightning-generated NO$_x$. The lines are linear least squares fits. The error bars correspond to a 1-sigma standard deviation.**

It is clear that there are greater changes when nitrate is considered compared to changes caused by changes in LNO$_x$ scheme.

We also examined the modelled tropospheric cloud droplet number concentration (CDNC) (one could also look at CCN or
condensation nuclei (CN), but such output was not available from the model runs made). CDNC can be used as a proxy for
CCN (the latter is a measure of the potential to form cloud droplets at the bottom of the cloud. Typical cloud droplet is 20
µm in diameter. Apart from modifying the aerosol size distribution, nitrate also modifies the chemical composition of the
aerosol which can change the CCN efficiency of the particles with ramifications for the indirect aerosol effects (Tost, 2017).
The incorporation of nitrate in the model causes an average increase of ∼ 3.5% in CDNC for any given LNO$_x$ value (Figure
15). The average increase in CDNC per Tg[N] yr$^{-1}$ of LNO$_x$ is 0.035 cm$^{-3}$. Compared to the no-LNOx case, there is a ∼ 3%

increase in the mean tropospheric CDNC when LNO$_x$ is considered (via the Lu21 scheme) (see Supplement S2 for zonal CDNC distributions).


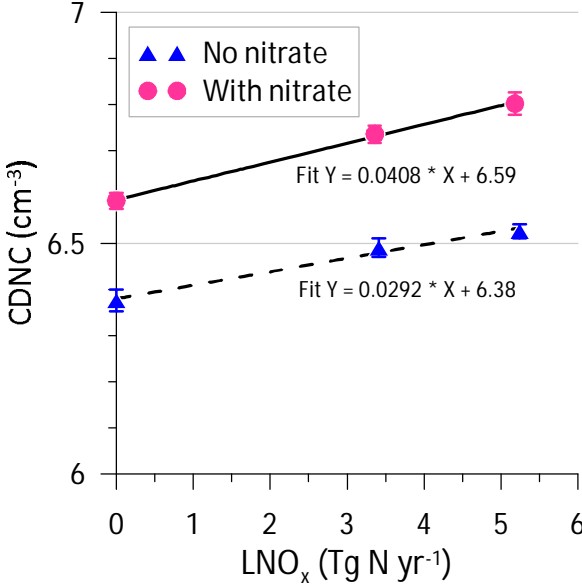

**Figure 15: Modelled annual-mean tropospheric cloud droplet number concentration (CDNC), as a function of lightning-generated NO$_x$. The lines are linear least squares fits. The error bars correspond to a 1-sigma standard deviation.**

In this work, more complex flash-rate schemes can also be tried, for example the upward cloud ice flux based method of Finney et al. (2016) which is shown to perform better in the subtropics and part of the midlatitudes compared to the cloud-top height based schemes used here. However, we believe that the results on globally averaged atmospheric composition changes with LNO$_x$ obtained here are not likely to change, but there could be regional impacts.

**5 Conclusions**

In this sensitivity focused study, we have essentially addressed two problems through the use of a global chemistry-climate model, UM-GA8.0-UKCA: 1) quantifying the impact of including nitrate aerosol on global mean quantities such as tropospheric composition, and 2) the dependency of these effects on lightning generated NO$_x$. The latter was explored by considering two empirical lightning flash-rate parameterisations: the PR92 scheme (Price and Rind, 1992) and the Lu21 scheme (Luhar et al., 2021), with the Lu21 scheme improving upon the underestimation of flash rate by the PR92 scheme

over the ocean. Apart from the changes in the various global mean quantities without and with nitrate, such changes per Tg[N] yr$^{-1}$ change in LNO$_x$ were also presented.

The amount of global LNO$_x$ obtained from the Lu21 scheme was about 50% higher than that from the PR92 scheme (5.2 vs. 3.4 Tg[N] yr$^{-1}$). This variation in LNO$_x$, together with the simulations with zero LNO$_x$, enabled an investigation of the change in various globally averaged modelled quantities as a function of LNO$_x$. We found that both nitrate aerosol and

changes in LNO$_x$ lead to significant changes in tropospheric composition and aerosol responses We also found that the difference between values of the various global mean quantities obtained without and with nitrate is almost constant regardless of the value of the total global LNO$_x$ emission used.

With the inclusion of nitrate aerosol, there was a decrease (of ~ 4–5%) in the mean tropospheric O$_3$ level, with the biggest reductions located in the mid to upper troposphere in the Northern Hemisphere. The methane lifetime increased by

approximately 5% (~ 0.4 years) as the mean tropospheric OH concentration decreased by a similar percentage. There were reductions in NH$_3$, HNO$_3$, gaseous nitrate radical and N$_2$O$_5$ mixing ratios. Aerosol size distribution also changed when nitrate aerosol is included. We found that there was a very small decrease of ~ 1% in aerosol number concentration in the nucleation mode, a reduction of ~ 6.4% in the Aitken mode, and an enhancement of ~ 10% in the accumulation mode. Of all the modes, aerosols in the accumulation mode are of most importance from climate impact, thus this change is significant.

The mean AOD$_{550}$ increased by 11%, the mean tropospheric CDNC by 3.5%, and the change in R$_n^{TOA}$ was ~ –0.4 W m$^{-2}$. These results build on those presented in Jones et al. (2021) by determining the impact of nitrate aerosol on tropospheric composition, and will be useful, for example, for the development of UKESM2.

Comparing simulations with and without LNO$_x$ emissions (corresponding to an LNO$_x$ difference of 5.2 Tg[N] yr$^{-1}$), we show that the impact of LNO$_x$ on global-mean tropospheric aerosol composition is an increase of 2.8% in NH$_4$, 4.7% in fine NO$_3$,

12% in coarse NO$_3$, and 5.8% in SO$_4$ by mass; a small 1% increase in AOD$_{550}$ and an increase of 3% in CDNC.

Moving from the PR92 to Lu21 scheme (both with nitrate included), there was a very small global mean increase (~ 1%) in both NH$_4$ and fine NO$_3$ aerosol mass burdens. On the other hand, the increase in the coarse NO$_3$ was greater (~ 4%). This change in LNO$_x$ scheme also increased the tropospheric SO$_4$ aerosol burden by ~ 1.7%. These aerosol changes could be ascribed to particular aerosol modes, and they had considerable regional variations. These increases were dominated by the

aerosol mass concentration in the lower troposphere. There was an increase of ~ 3.6% in the aerosol number concentration in the nucleation mode and an increase of ~ 5.6% in the Aitken mode, and these increases were dominated by increases in the mid to upper troposphere. A small decrease of ~ 1.5% was estimated in the accumulation mode with negligible changes in the other modes. Compared to the PR92 scheme, the Lu21 scheme yielded a very small (0.5%) overall increase in the global AOD$_{550}$. The mean R$_n^{TOA}$ increased by ~ 0.07 W m$^{-2}$, which suggests that the positive radiative feedback from an increase in

ozone dominates over negative radiative feedback resulting from a reduction in methane lifetime and increase in aerosol concentration as LNO$_x$ is increased in the Lu21 scheme. In general, we find that the magnitude of changes in gas-phase

tropospheric composition as a result of changes in $LNO_x$ going from the PR92 to Lu21 scheme is roughly comparable to that caused by the inclusion of nitrate aerosol. But the magnitude of changes in $AOD_{550}$, $R_n^{TOA}$ and CDNC are dominated by inclusion of nitrate.

We assumed a single value for the $HNO_3$ uptake coefficient ($\gamma$) to produce $NH_4NO_3$, corresponding to the FAST value (= 0.193) used in Jones et al (2021) which likely represents an upper limit on nitrate effects. Given uncertainties over the composition, relative-humidity and temperature dependence of the $HNO_3$ uptake coefficient, it is not yet possible to perform a more comprehensive study in which the uptake rate is a dependent on these variables. Any direct new particle formation (due to lightning or otherwise) is not yet explicitly included in the nitrate scheme in UM-UKCA, which could also be

important – this may be considered in future versions of the UM. The results obtained here on the degree of sensitivity to nitrate aerosol and $LNO_x$ will be useful for further $LNO_x$ and nitrate impacts assessment in future UM studies. Nitrate concentrations are sensitive to precursor emissions, and we used constant emissions forcings representative of the year 2000 following CMIP6 protocol. A more comprehensive study could include transient emissions to investigate recent trends in ammonium nitrate concentrations.

In this paper, we have shown that simulating nitrate in GCMs is important for tropospheric composition, alongside radiation and cloud droplet activation. We have also shown that simulating $LNO_x$ production over the ocean (as in Lu21) produces a tangible impact on regional aerosol concentrations at the surface, though the largest impacts are at higher altitudes where $LNO_x$ and $NH_3$ coexist. Our results could be used to infer the impact of changing lightning rates (and $LNO_x$ emissions) on nitrate concentrations under climate change.

**Code availability**

Owing to intellectual property rights restrictions, we cannot provide the source code or documentation papers for the UM. The Met Office UM is available for use under licence. A number of research organisations and national meteorological services use the UM in collaboration with the Met Office to undertake basic atmospheric process research, produce forecasts, and develop the UM code. To apply for a licence, see http://www.metoffice.gov.uk/research/modelling-systems/unified-

model (Met Office, 2024). Geospatial figures were produced using Python 3.10.8 (https://www.python.org, last access: 27 February 2024) and Iris 2.4.0 (https://scitools.org.uk, last access: 27 February 2024).

**Data availability**

Output from the model suites used here is available on MASS tape archive and can be accessed upon request by obtaining a login on U.K.'s environmental science data analysis facility at http://www.jasmin.ac.uk. The model suites used UM@vn13.2

and the suite IDs corresponding to the simulation names NN_0, NN_PR, NN_Lu, WN_0, WN_PR and WN_Lu used in this

paper are u-cx746, u-cx815, u-cx814, u-cx745, u-cx817 and u-cx819, respectively. The LIS/OTD lightning flash data (V2.3.2015) were available from https://lightning.nsstc.nasa.gov/data/data_lis-otd-climatology.html (last access: 6 May 2021).

**Author contributions**

AKL designed the study, setup the modelling experiments, developed analysis scripts, analysed model output, and prepared the paper with assistance from all co-authors. ACJ contributed to setting up the modelling experiments, advised on aerosol in the model, oversaw the progress of simulations, and assisted with the development of analysis codes. JMW assisted with and carried out part of the modelling work pertaining to lightning schemes.

**Competing interests**

The authors declare that they have no conflict of interest.

**Acknowledgements**

This work was supported by the CSIRO-UM Partnership Agreement project with the Met Office (CSIRO projects OD-207701, OD-236140). Ashok Luhar acknowledges a travel award from the UM Partner Visiting Scientist Exchange (UNITE) Program to visit the Met Office, Exeter, in June-July 20203 and thanks the UNITE Program manager Joao Teixeira for the
assistance provided with the application process and during the visit. We acknowledge Luke Roberts of the Met Office for his assistance with setting up computational environment on JASMIN (U.K.'s storage and compute facility for environmental sciences data analysis) to carry out model output analysis, and Martin Cope of CSIRO for his helpful comments on this work. We thank the two anonymous reviewers for their useful comments

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
