# Peer review of "Quantifying the impact of global nitrate aerosol on tropospheric composition fields and its production from lightning NOx"

_EGUsphere, 2024_

## Author Comment (AC1)

**Reply by the authors to Referee #1's comments on**
**"Quantifying the impact of global nitrate aerosol on tropospheric composition fields and its production from lightning NOx" (https://doi.org/10.5194/egusphere-2024-1363-RC1)**

**Anonymous Referee #1 (RC1)**

We are grateful to the Referee for giving their time to provide a thorough review of our manuscript and making a number of helpful comments. In the following, we provide our responses to these comments (the Referee's comments are shown in blue).

**General Remarks:**

The authors studied the contribution of lightning nitrogen oxides (LNOx) to global nitrate aerosols and the impact of global nitrate aerosols on tropospheric composition, aerosol optical depth (AOD), and atmospheric radiation fields. They found that lightning leads to an increase in atmospheric nitrate and sulfate aerosols, with the most significant increase in coarse-mode nitrate, up to 12%. The inclusion of nitrate aerosols reduced tropospheric ozone and increased methane lifetime, both by about 4-5%. The reduction in atmospheric oxidants caused by the inclusion of nitrates is one of the reasons. They also reported that nitrate aerosols increase AOD and contribute -0.4 W m-2 to the net downward radiation flux at the top of the atmosphere. This is an interesting and valuable study. I recommend it for publication in ACP after the authors make the following minor revisions.

**Response:** We are glad you found this study interesting and valuable.

**Major comments:**

The main concern of this study is how confident one can be about the reported contribution of nitrate to the atmospheric chemical and radiative fields. The HNO3 absorption rate is a key factor that directly affects nitrate formation and its subsequent contribution. The rates reported in this study range from 0.193 (fast rate, used in this study) to 0.001 (slow rate). The uptake rates vary by more than a factor of one thousand, so further discussion of the uncertainties in the nitrate effect is necessary, at least qualitatively. For example, the authors may need to conduct a literature review summarizing HNO3 uptake rates on various aerosol types measured in the fields and in the laboratories. They can further provide qualitative uncertainty estimates by combining the measured uptake rates with nitrate formation from various aerosol components.

**Response:** The Referee is right to highlight that the uncertainty in $HNO_3$ uptake rate may vary with aerosol composition and that this will have implications for the resultant concentration of nitrate aerosol. We conducted a thorough literature review in Jones et al (2021) and found significant uncertainty in the composition-dependence of the $HNO_3$ uptake rate. In that study, we decided to test values at either end of the spectrum from our literature review. The value of 0.193, which we use in this report, was shown to produce similar results globally to instantaneous thermodynamic equilibrium, in line with other models that make this assumption, which justifies its usage here. In Section 3.1, we already have this caveat explaining uncertainty over the $HNO_3$ uptake coefficient, and we now also discuss the sensitivity of the results to the uptake rate coefficient when it is changed from 0.193 (fast rate, used in this study) to 0.001 (slow rate):

"Jones et al. (2021) tested the sensitivity of $NH_4NO_3$ aerosol concentrations to the $HNO_3$ uptake coefficient for the $NH_3$-$HNO_3$ uptake on Aitken and accumulation soluble particles (Table 1) with

two values selected from the literature, $\gamma = 0.193$ (FAST) and 0.001 (SLOW), representing fast and slow uptake rates, respectively. They found that, generally, the fast uptake value shows a higher spatial correlation with measured nitrate surface concentrations whereas the slow value simulates their magnitudes better. They also found that compared to FAST, the SLOW value led to a 58% and 52% reduction in the global near-surface concentration and burden of fine particulate nitrate, respectively. The reductions in $NH_4$ were 24% and 15%, while coarse mode $NO_3$ remained almost unchanged. Aerosol optical depth decreased by 6%, and the magnitude of the TOA net downward radiative flux changed by 63%. This sensitivity test showed that despite a two-hundredfold variation in the uptake rate, the model's response was nonlinear and perhaps less sensitive than expected. In this study, we use the FAST value $\gamma = 0.193$, which is currently the default in UKCA-mode. Jones et al (2021) showed that this value produces similar results globally to the widely utilised assumption of instantaneous thermodynamic equilibrium. This suggests that our results likely represent the upper end of efficiency of $NH_4$ and $NO_3$ production and its impact in the UM. Jones et al. (2021) recognised that rather than being globally invariant, $\gamma$ may vary with aerosol composition, temperature, and relativity humidity, and needs better constraining, thus needing further research and future model development outside the scope of the present study."

Additionally, we include the following caveat in Conclusions:

"We have assumed a single value for the $HNO_3$ uptake coefficient ($\gamma$), corresponding to the FAST value (0.193) used in Jones et al (2021) which likely represents an upper limit on nitrate effects. Given uncertainties over the composition, relative-humidity and temperature dependence of the $HNO_3$ uptake coefficient, it is not yet possible to perform a more comprehensive study in which the uptake rate is a dependent on these variables."

The main scientific goal of this study is to investigate nitrate-related atmospheric effects, so the description of the chemical mechanism of nitrate formation needs to be strengthened. What features of the formation of fine-mode NH4NO3 component make it a quasi-instantaneous thermodynamic equilibrium scheme? How does this approach differ from the "commonly used instantaneous thermodynamic equilibrium nitrate scheme"? What are the advantages of using a quasi-instantaneous thermodynamic equilibrium scheme in this study? The authors should make it clear in the abstract that their results likely represent an upper limit on nitrate effects, as they used a fast uptake rate for the condensation of HNO3 to produce NH4NO3.

**Response:** While the chemical mechanism of nitrate formation is fully described in Jones et al. (2021) and in its supplement, we agree with the Referee that its description in the present paper needs to be strengthened. To address this, we add the following text.

We expand on the text to read

"The component of the new nitrate scheme dealing with the formation of fine-mode $NH_4NO_3$ from the condensation of $HNO_3$ and $NH_3$ is numerically solved first, prior to the condensation of $HNO_3$ on coarse aerosols (i.e., dust and sea salt). Most fine-mode nitrate schemes assume that $NH_4NO_3$ concentrations reach thermodynamic equilibrium instantaneously, without accounting for the kinetic limitations on the condensation of $HNO_3$ or $NH_3$ onto existing aerosol particles. Instead, our quasi-instantaneous thermodynamic equilibrium scheme assumes an exponential decay of the gas phase toward equilibrium, using an equilibration time scale ($\tau_e$). This approach is based on Schwartz's (1986) first-order uptake theory and incorporates correction factors from Fuchs and Sutugin (1970) to account for molecular effects and limitations in interfacial mass transport. $\tau_e$ is a function of the $HNO_3$ condensation or uptake rate coefficient ($\gamma$), a key parameter in the first-order

uptake theory and defined as the number of gas molecules condensing on a particle divided by the number impacting onto the particle surface. The higher the uptake coefficient the smaller the equilibration time scale. The benefit of using such a scheme is that it realistically constrains the rate at which $NH_4NO_3$ concentrations achieve equilibrium."

We now state in the abstract that "The results likely represent an upper limit on nitrate effects, as they were derived using a fast uptake rate for the condensation of $HNO_3$ to produce $NH_4NO_3$".

The authors should also elaborate on how the model treats coarse-mode nitrate formation. Does the model use first-order condensation of HNO3 on dust and sea salt? If so, what are the corresponding uptake rates?

**Response:** In Section 3.1, we add "In our nitrate scheme, coarse nitrate is present in the accumulation and coarse soluble modes. Following $NH_4NO_3$ production and the associated update to $HNO_3$ concentrations, the first-order uptake parameterisation is further employed to model the irreversible uptake of $HNO_3$ on sea salt and dust to produce $NaNO_3$ and $Ca(NO_3)_2$, respectively. This reaction is slower than ammonium nitrate production, therefore numerically ammonium nitrate production is solved first. The $HNO_3$ uptake coefficients for CLASSIC dust and sea salt are relative humidity dependent variables based on measurements from Fairlie et al. (2010) and Sander et al. (2011), respectively. Dust is assumed to uniformly constitute 5% $Ca^{2+}$ by mass (Jones et al., 2021)."

It is strongly recommended that the authors add tables summarizing their findings on changes in atmospheric composition, AOD, and radiation fields due to lightning and the presence of nitrates. These results in the current framework are scattered throughout the tests and are difficult for the readers to follow. At the same time, the authors should also summarize the emissions from various nitrogen emission types in a table to help readers understand the relative importance of lightning and other emissions to the atmospheric chemistry and radiation fields. It would also be helpful for the authors to report the atmospheric oxidant fields and their changes from the designed experiments in tabular form, as changes in the oxidant field are one of the key reasons for the corresponding atmospheric composition changes.

**Response:** We agree with the Referee. We have now prepared a table (Table 4, as below) giving changes in global-mean atmospheric composition, OH, AOD, radiation and other parameters due to lightning and the presence of nitrate. What is important in this Table is the relative change in the various parameter values when nitrate is included and how these parameter values change per Tg N yr$^{-1}$ of $LNO_x$ (based on the slopes of linear least-squares fits). The emissions from various nitrogen emission types are given in Table 3 (as below).

**Table 3: Annual totals of global emissions of various nitrogen types prescribed in the UM-UKCA simulations.**

| Species | Source | Emissions (Tg N yr$^{-1}$) |
|---|---|---|
| $NH_3$ | Oceanic | 8.1 |
| $NH_3$ | Anthropogenic | 41.6 |
| $NH_3$ | Biomass | 3.9 |
| $NO_x$ | Soil | 5.5 |
| $NO_x$ | Anthropogenic | 35.8 |
| $NO_x$ | Biomass | 7.5 |
| $NO_x$ | Aircraft | 0.7 |

**Table 4: Modelled global averages of various atmospheric variables obtained from the no-nitrate (NN) and with-nitrate (WN) simulations for three lightning $NO_x$ setup options: no $LNO_x$, Price and Rind's (1992) (PR92 or PR) lightning scheme and Luhar et al. (2021) (Lu21 or Lu) lightning scheme. All species are tropospheric averages, aerosol no. is aerosol number concentration, nu = nucleation mode, Ai = Aitken mode, Ai (in) = Aitken insoluble mode, ac = accumulation mode, co = coarse mode.**

| Simulation | Global variable | Lightning scheme | | | Change per Tg N yr$^{-1}$ of $LNO_x$ |
|---|---|---|---|---|---|
| | | None | PR92 | Lu21 | |
| No nitrate | $LNO_x$ emission (Tg N yr$^{-1}$) | 0 | 3.41 | 5.24 | - |
| | $O_3$ burden (Tg) | 278.5 | 325.2 | 348.8 | 13.45 |
| | $O_3$ (ppbv) | 48.1 | 56.7 | 60.9 | 2.46 |
| | OH ($\times 10^5$ molec. cm$^{-3}$) | 8.95 | 11.32 | 12.63 | 0.70 |
| | Methane lifetime (yr) | 8.55 | 7.49 | 7.03 | -0.35 |
| | CO (ppbv) | 93.7 | 82.4 | 78.1 | -3.03 |
| | NO (pptv) | 14.2 | 20.1 | 23.9 | 1.85 |
| | $NO_2$ (pptv) | 36.4 | 45.7 | 51.6 | 2.89 |
| | $NH_3$ (pptv) | 177.0 | 176.0 | 176.5 | -0.12 |
| | $HNO_3$ (pptv) | 151.0 | 190.2 | 213.8 | 11.9 |
| | $N_2O_5$ (pptv) | 0.41 | 0.63 | 0.78 | 0.070 |
| | $NO_3$ radical (pptv) | 0.37 | 0.45 | 0.50 | 0.023 |
| | $SO_4$ burden (µg[S] m$^{-2}$) | 981.5 | 1019.8 | 1036.2 | 10.54 |
| | Aerosol no.: nu (cm$^{-3}$) | 1749.8 | 1821.4 | 1890.8 | 26.16 |
| | Aerosol no.: Ai (cm$^{-3}$) | 331.2 | 360.0 | 377.8 | 8.84 |
| | Aerosol no.: Ai (in) (cm$^{-3}$) | 8.64 | 8.70 | 8.67 | 0.0065 |
| | Aerosol no.: ac (cm$^{-3}$) | 39.1 | 40.7 | 41.4 | 0.510 |
| | Aerosol no.: co (cm$^{-3}$) | 0.154 | 0.155 | 0.155 | 0.0002 |
| | AOD | 0.1393 | 0.1407 | 0.1410 | 0.00034 |
| | $R_n^{TOA}$ (W m$^{-2}$) | 0.50 | 0.64 | 0.71 | 0.040 |
| | CDNC (cm$^{-3}$) | 6.38 | 6.49 | 6.53 | 0.029 |

| | | | | | |
|---|---|---|---|---|---|
| | $LNO_x$ emission (Tg N $yr^{-1}$) | 0 | 3.36 | 5.18 | - |
| | $O_3$ burden (Tg) | 260.8 | 307.1 | 332.6 | 13.85 |
| | $O_3$ (ppbv) | 44.9 | 53.5 | 58.1 | 2.55 |
| | OH ($\times 10^5$ molec. $cm^{-3}$) | 8.20 | 10.61 | 12.01 | 0.73 |
| | Methane lifetime (yr) | 9.15 | 7.90 | 7.38 | -0.29 |
| | CO (ppbv) | 98.5 | 85.4 | 80.4 | -3.55 |
| | NO (pptv) | 12.7 | 18.1 | 22.1 | 1.79 |
| | $NO_2$ (pptv) | 33.5 | 42.1 | 48.2 | 2.80 |
| | $NH_3$ (pptv) | 29.8 | 27.9 | 27.2 | -0.51 |
| | $HNO_3$ (pptv) | 87.9 | 122.2 | 144.0 | 10.8 |
| | $N_2O_5$ (pptv) | 0.31 | 0.51 | 0.65 | 0.065 |
| | $NO_3$ radical (pptv) | 0.32 | 0.39 | 0.44 | 0.023 |
| With nitrate | $SO_4$ burden ($\mu g[S]$ $m^{-2}$) | 995.1 | 1035.5 | 1052.7 | 11.23 |
| | $NH_4$ burden ($\mu g[N]$ $m^{-2}$) | 814.3 | 831.3 | 837.0 | 4.47 |
| | Fine $NO_3$ burden ($\mu g[N]$ $m^{-2}$) | 291.3 | 301.9 | 304.8 | 2.68 |
| | Coarse $NO_3$ burden ($\mu g[N]$ $m^{-2}$) | 124.7 | 134.5 | 139.7 | 2.90 |
| | Aerosol no.: nu ($cm^{-3}$) | 1721.1 | 1807.5 | 1875.1 | 29.23 |
| | Aerosol no.: Ai ($cm^{-3}$) | 302.0 | 333.9 | 353.6 | 9.90 |
| | Aerosol no.: Ai (in) ($cm^{-3}$) | 9.10 | 9.08 | 9.13 | 0.0043 |
| | Aerosol no.: ac ($cm^{-3}$) | 42.9 | 44.8 | 45.5 | 0.443 |
| | Aerosol no.: co ($cm^{-3}$) | 0.163 | 0.166 | 0.166 | 0.00062 |
| | AOD | 0.1539 | 0.1546 | 0.1553 | 0.00026 |
| | $R_n^{TOA}$ (W $m^{-2}$) | 0.11 | 0.25 | 0.29 | 0.036 |
| | CDNC ($cm^{-3}$) | 6.59 | 6.73 | 6.80 | 0.041 |

**Specific comments:**

1. The conclusions in P1L19-25 are controversial. P1L19-23 showed that the inclusion of LNOx increases atmospheric nitrate and sulfate aerosols, and that the inclusion of nitrate aerosols leads to a reduction in tropospheric ozone loading. This effect of reduced tropospheric ozone appears to be in dispute with P1L24-25, which showed that with increased LNOx, global AOD and top-of-atmosphere net downward radiative flux increase through increased tropospheric ozone.

**Response:** It seems there is some confusion here caused by a lack of clarity. Here we are talking about the impact of two separate things: inclusion of nitrate and an increase in $LNO_x$. To improve the clarity, we have revised the text as follows:

"With the inclusion of nitrate aerosol, the global mean tropospheric OH level decreases by 5%, the tropospheric ozone burden drops by 4–5%, the tropospheric methane lifetime increases by a similar magnitude, and the top-of-atmosphere (TOA) net downward radiative flux changes by $–0.4$ W $m^{-2}$. An increase of 5.2 Tg N $yr^{-1}$ in LNOx from a zero baseline leads to a global mean increase of 2.8% in $NH_4$, 4.7% in fine $NO_3$, 12% in coarse $NO_3$, and 5.8% in $SO_4$ aerosol mass burdens, showing that $LNO_x$ has a greater impact on coarse aerosol. The inclusion of nitrate aerosol also shifts the aerosol size distribution, with the most notable changes in the Aitken and accumulation modes. Regardless of nitrate aerosol inclusion, increasing $LNO_x$ results in relatively small positive enhancements in the global mean AOD and TOA net downward radiative flux (with the latter change dominated by an increase in tropospheric ozone as $LNO_x$ increases)."

2. P3L87: What are these "conductive atmospheric conditions"?

**Response:** We now elaborate on these conditions by saying "To give an example, whilst $NO_x$ emissions from lightning are comparable in magnitude to those from soils or biomass burning, they contribute about three times as much to the total tropospheric $O_3$ column (Dahlmann et al., 2011). This is because, in the middle to upper troposphere where lightning $NO_x$ is released, the $O_3$ production efficiency per unit of $NO_x$ is significantly higher ($\sim 100$ molecules of $O_3$ per molecule of $NO_x$) compared to near the surface ($\sim 10$–$30$ molecules of $O_3$ per molecule of $NO_x$) due to the higher amount of UV radiance, lower concentrations and longer lifetimes of $NO_x$ (days instead of hours), and cooler temperatures affecting ozone loss chemistry at such altitudes (Dahlmann et al., 2011)."

3. P5L142: Can you give the mean global ocean flash rate based on Eq. (3) and the observations?

**Response:** Done. We now say "… predicts a mean global flash rate that is smaller by a factor of approximately 30 compared to the observed (a predicted global oceanic average of 0.33 flashes $s^{-1}$ compared to the observed 9.16 flashes $s^{-1}$) …"

4. P6L158 and P6L166: Given Pno, ic = Pno, cg, how do you partition the flash rate (Fl or Fo) into Fic and Fcg?

**Response:** We now add "The fraction of CG lightning flashes is determined based on cold cloud thickness, following an empirical relationship developed by Price and Rind (1993), where cold cloud thickness is further parameterised as a function of latitude. The remaining fraction is then equal to the IC flash fraction. These fractions multiplied with the calculated flash rate ($F_L$ or $F_O$) give $F_{CG}$ and $F_{IC}$, respectively. The calculated NO at a specific location and time step is distributed vertically in the grid column using a linear distribution in log(pressure) coordinates. For IC flashes, this extends from 500 hPa to the cloud top, and for CG flashes, from 500 hPa to the surface (Archibald et al., 2020; Luhar et al., 2021)."

5. P8L226: For which aerosol particles does the HNO3 uptake rate coefficient apply here?

**Response:** The FAST $HNO_3$ uptake coefficient applies to $NH_3$-$HNO_3$ uptake on Aitken and accumulation soluble particles. This is now clarified.

6. P12L334: Please explain the significance level defined here? How was the significance level calculated?

**Response:** Annually averaged data from our 15-year model simulation (sample size = 15) were used in a t-test to determine whether the means of two populations differed significantly at a 95% confidence level. This is now stated.

7. Figure 8: The figure only shows the difference between the two simulations (Lu21 – no LNOx), instead of "Annual-mean tropospheric NH4, fine NO3, coarse NO3, and SO4 burdens from the Lu21 and no-LNOx simulations (both with nitrate) and the differences …." in the figure caption.

**Response:** These are indeed only the difference plots. The caption has been fixed.

8. P20L484: Why is "Conversely…." used here? It would be better to explain this sentence and the previous one in more detail.

**Response:** We add "…due to the much greater concentrations of $NH_3$ from agricultural sources."

9. P21L501: How much have tropospheric oxidants increased?

**Response:** We modify the sentence to read "The overall increase in $SO_4$ with the Lu21 scheme is possibly due to increase in tropospheric oxidants in response to increase in $LNO_x$ (for example, there is a 13% increase in OH with the Lu21 scheme compared to the PR92 scheme with nitrate included)."

10. P23L519-521: "The coarse mode (soluble) particles (plot not shown) are the least in number and confined to very close to the surface due to their effective gravitational sedimentation, with more particles in the Southern Hemisphere than in the Northern Hemisphere (probably due to a larger oceanic surface in the NH so as to cause a large sea salt particle number concentration)." Please check the sentence. The oceanic surface in the NH is smaller than in the SH.

**Response:** The text has been fixed (NH changed to SH).

11. P25L507: How to obtain the uncertainty AOD here?

**Response:** We believe the Referees is pointing to P27L607. We modify the sentence to "The 1-sigma standard deviation uncertainty bars (calculated from the annual means over 15 years of simulation) …"

12. P28L620-622: It is suggested that the sentence be changed to "The negative changes in RnTOA in simulations with and without nitrate indicate a reduction in atmospheric radiation absorption, implying cooling conditions when nitrate is considered." Such a statement is more in line with the common sense of the aerosol cooling effect.

**Response:** Point taken. We modify the text to "A decrease or negative change in $R_n^{TOA}$ signifies reduced atmospheric radiation absorption, indicating cooling conditions. In Figure 14, $R_n^{TOA}$ increases with $LNO_x$ which suggests that the positive radiative feedback from ozone increases outweighs the negative feedback from the reduction in methane lifetime and rise in aerosol concentrations as $LNO_x$ increases. This holds true whether nitrate is included or not. However, the inclusion of nitrate results in a change of approximately $-0.4$ $W\ m^{-2}$ in $R_n^{TOA}$ for any given $LNO_x$ level (Figure 14), indicating that the incorporation of nitrate in the model has a far greater impact on $R_n^{TOA}$ than the change in $LNO_x$ considered here."

13. P30L642-643: What is the difference in tropospheric CDNC with and without nitrate included?

**Response:** We add "The incorporation of nitrate in the model causes an increase of 4.2% in CDNC".

14. P31L656: What are these "globally averaged modelled atmospheric parameters"?

**Response:** We modify the sentence to "This variation in $LNO_x$, together with the simulations with zero $LNO_x$, enabled an investigation of the change in globally averaged modelled properties such as the tropospheric ozone burden, methane lifetime, OH concentration, AOD and $R_n^{TOA}$ as a function of $LNO_x$."

15. P31L660: Suggest further analysis of changes in the atmospheric oxidant field.

**Response:** We modify the sentence to "The methane lifetime increased by approximately 5% (~ 0.4 years) as the mean tropospheric OH concentration decreased by a similar percentage."

16. P31L666-667: Please elaborate on "the variation considered in LNOx". What is the variation you are referring here?

**Response:** We change the sentence to "However, the change in $R_n^{TOA}$ when nitrate is included was ~ – 0.4 W m$^{-2}$ which suggests that incorporation of nitrate in the model has a much bigger impact on $R_n^{TOA}$ than the magnitude of change in this quantity when $LNO_x$ is varied from 0 to 5.2 Tg N yr$^{-1}$."

**Technique corrections:**

P2L47: Delete "production" after oxidation.

**Response:** Done.

P4L124: Change the sentence to be "this is followed by results and discussion in Section 4 and then conclusions in Section 5".

**Response:** Done.

P6L152: Change "parameterisation" to "parameterization".

**Response:** We have followed British English, and use "parameterisation".

P7L193-199: To make the logic more reasonable, move the sentence "Radiative changes include direct aerosol radiative forcing …." before the sentence "The Predicted Cloud Cover and Predicted Condensate (PC2) schemes ….".

**Response:** Done.

P8L213: What is the "10" in "the UKLA-model setup 10"?

**Response:** UKCA-mode can be run with several possible aerosol configurations (referred to as "mode setups". Numbers are assigned to different model setups just as a reference. The setup 10 used in our paper corresponds to the options in Table 1 with the nitrate scheme and the 6-bin CLASSIC dust scheme. We change the phrase "this UKCA-mode setup together with the CLASSIC dust scheme is referred to as setup 10".

P17L437-438: Change "particularly over South Asia with values as high as 3 mg[N] m-2 and over East Asia / China with values as high as 2 mg[N] m-2. Over central North America, values as high as 1.1 mg[N] m-2 are predicted" to "particularly the values as high as 3 mg[N] m-2 in South Asia, 2 mg[N] m-2 in East Asia/China, and 1.1 mg[N] m-2 in central North America."

**Response:** Done.

---

## Author Comment (AC2)

**Reply by the authors to Referee #2's comments on**
**"Quantifying the impact of global nitrate aerosol on tropospheric composition fields and its production from lightning NOx" ([https://doi.org/10.5194/egusphere-2024-1363-RC2](https://doi.org/10.5194/egusphere-2024-1363-RC2))**

**Anonymous Referee #2 (RC2)**

We are grateful to the Referee for their time to review our manuscript and making a number of important points. In the following, we provide our responses to these comments (the Referee's comments are shown in blue).

**Summary**

This manuscript uses the UKCA chemistry-climate model (CCM) to evaluate the impact of lighting NOx emissions on particulate nitrate chemistry (recently implemented in the UKCA model), and thereby, its influence on climate.

Overall Comment

I see no overall flaws with the study as performed and reported.

**Response:** Thank you.

However, the manuscript as written oversells its originality and its accuracy. Many global atmospheric chemistry models — including those cited in the introduction — have had nitrate aerosol chemistry for decades, e.g., GEOS-Chem (Park et al., 2004), GISS (Bauer et al., 2007), GFDL (Paulot et al., 2016; and ref. therein).

**Response:** Our intention was not to oversell the paper's originality or accuracy, and we now add suitable caveats and references as suggested by the Referee.

In Introduction when we say "*Despite the above, few global chemistry-climate models include nitrate aerosol, and usually its effects are completely ignored (Tost, 2017), with the main reason being the chemical complexity of nitrate formation and the semi-volatile nature of ammonium nitrate. To give an example, of the ten global Earth system models that participated in the Aerosol and Chemistry Model Intercomparison Project (AerChemMIP) of the Coupled Model Intercomparison Project Phase 6 (CMIP6), aimed at understanding the effects of reactive gases and aerosols on Earth's climate, only two had an interactive stratospheric and tropospheric gas-phase and aerosol- chemistry scheme together with an explicit treatment of nitrate aerosol (Thornhill et al., 2021)*" those two models are indeed GISS and GFDL (GEOS-Chem is a chemical transport model, not a chemistry-climate model, so not included). We have revised the text to read:

"Although nitrate aerosol has been included in some global models, such as the chemical transport model GEOS-Chem (e.g., Park et al., 2004) and the chemistry-climate models GISS (e.g., Bauer et al., 2007) and GFDL (e.g., Paulot et al., 2016), it is often ignored in global chemistry-climate models (Tost, 2017). This may be partly due to the computational cost of simulating nitrate, combined with the chemical complexity of its formation and the semi-volatile nature of ammonium nitrate, which can reevaporate into the atmosphere (e.g., Stelson et al., 1979). In fact, out of the ten global Earth system models with atmospheric chemistry that participated in the Aerosol and

Chemistry Model Intercomparison Project (AerChemMIP) under the Coupled Model Intercomparison Project Phase 6 (CMIP6), which aims to assess the effects of reactive gases and aerosols on Earth's climate, only the GISS and GFDL models explicitly treated nitrate aerosol along with an interactive stratospheric and tropospheric chemistry scheme (Thornhill et al., 2021)."

Therefore, any of the many publications that have used these models to look at the impact of lightning on atmospheric composition and climate have included the impact of nitrate particles. The main reason that it has not been highlighted in the manuscripts is because it was noticed be negligible compared to other impacts and/or the nitrate simulation was evaluated to have too poor a skill in reproducing observational constraints, precluding meaningful conclusions.

**Response:** We believe all current global chemistry-climate models include lightning-generated emissions of $NO_x$ ($LNO_x$) irrespective of whether nitrate aerosol is included or not. The lightning-nitrate relationship exists at least implicitly in studies that have used the models listed by the Referee, even if it has not been explored or quantified.

From the Referee's point of view, it may well have been the case that the impact of $LNO_x$ on nitrate aerosol was noticed to be negligible compared to anthropogenic sources or compared to other impacts (which we assume is what the Referee is stating), but the Referee does not point out any references that support their assertion that "it has not been highlighted … because it was noticed be negligible" nor their assertion about "poor a skill".  We would gladly cite any references that attest to either of these assertions, but the Referee has not provided references.

We have not seen any modelling study reported in the peer-reviewed literature that investigates the impact (significant or otherwise) of lightning on nitrate concentrations, other than that by Tost (2017). Similarly, we have not come across any study that examines how the modelled tropospheric composition is impacted when nitrate is accounted for.

We modify the relevant paragraph in Introduction to read:

"The area of quantifying the role of $LNO_x$ production on aerosol, particularly with nitrate aerosol included, has only received very limited attention compared to its role on gaseous atmospheric composition, and this could be due to reasons such as the inference that the low $LNO_x$ emission ($\sim 12\%$) compared to anthropogenic and biomass burning $NO_x$ sources contributes negligibly to nitrate concentrations. To our knowledge, the global modelling by Tost (2017), which involves a modal aerosol scheme with nitrate included, is the only study to explicitly examine the impact of $LNO_x$ on aerosol. It shows that $LNO_x$ (parameterised via the Price and Rind (1992) (PR92) scheme described below) is a significant source of nitrate in the upper troposphere and influences the aerosol size distribution and radiation. It is reported that chemical conversion of $LNO_x$ into $HNO_3$ is more favourable in the middle to upper troposphere, where lightning $NO_x$ mostly occurs, as compared to within the atmospheric boundary layer (where the dominant $NO_x$ and $NH_3$ sources are located) due to differences in chemical composition, chemical reactivity, and loss processes (Tost, 2017). Tost (2017) points to observational support for the occurrence of both $NH_3$ and $NO_3$ aerosol in convective outflows so that the formation of $NH_4NO_3$ is likely because of the low temperatures in the upper troposphere. Therefore, $LNO_x$ can change the spatial distribution of nitrate concentrations and concomitant climate impacts. Given the emerging importance of nitrate as sulfate concentrations wane it is important to assess the relative importance of all nitrate sources. At the same time, how modelled global atmospheric composition is impacted when nitrate is accounted needs to be quantified."

The UKCA implementation of nitrate as reported in Jones et al. (2021) similarly shows very poor skill in reproducing surface nitrate observations over the United States and Europe.

**Response:** The UKCA nitrate scheme compares very well to other AeroCom models and Earth system models (e.g. Table 3, Jones et al, 2021) on a global basis, and showed significant skill over the United States (e.g., R = 0.92 and 0.43 for $NH_4$ and $NO_3$, respectively, Figure 5, Jones et al., 2021), although the Referee is correct that the skill over Europe was at the time low. Jones et al (2021) attributed this to uncertainties in the $NH_3$ emissions inventory, as also found by Drugé et al. (2019) (cited in our paper). Recent tests with UKCA nitrate in the Met Office's AQUM (Air Quality in the Unified Model) have shown significant skill in predicting air quality episodes (https://www.ukca.ac.uk/images/5/59/PA_UKCA_Dec2022.pdf). In short, we have no reservations over the UKCA nitrate scheme.

I would support this manuscript's publication if it is recast as a theoretical exercise assuming the chemistry is correct (the lightning chemistry-climate interactions via nitrate and cloud microphysics are very interesting!) or if a proper evaluation of the in situ nitrate and aerosol optical depths were included to provide confidence in the simulation.

**Response:** In this paper, we use a state-of-the-art GCM with comprehensive stratospheric and tropospheric chemistry (StratTrop 1.0, used in UKESM2, Archibald et al., 2020 (cited)) as outlined in Section 3, with chemical emissions following the CMIP6 protocol, i.e., based on observations. We believe, it is incorrect therefore to describe this paper as a theoretical exercise when it is clearly designed to be as close to reality as possible. The only "theoretical" aspect of the methodology is the use of constant year 2000 conditions, which is used to remove the transient aspect of the emissions over the 20-year simulation period whilst maintaining a realistic geospatial emissions inventory. Rather, our paper can be considered as a sensitivity study. The results provide an important and useful quantification of the variation of the various fields considered with respect to the variation in $LNO_x$ and to whether or not nitrate is considered.

We add the following caveat to Conclusions:

"Nitrate concentrations are sensitive to precursor emissions, and in this study, we used constant emissions forcings representative of the year 2000 following CMIP6 protocol. A more comprehensive study could include transient emissions to investigate recent trends in ammonium nitrate concentrations."

We add the following to Introduction:

"Our modelling can be considered as a study of sensitivity of global fields of interest to changes in lightning $NO_x$ without and with nitrate aerosol."

And in Conclusions we already state:

"In this sensitivity focused study, we have essentially addressed two problems through the use of a global chemistry-climate model, UM-GA8.0-UKCA: 1) quantifying the impact of including nitrate aerosol on tropospheric composition, AOD and radiation, and 2) the dependency of these effects on lightning generated $NO_x$."

A comparison with MODIS derived AOD has been done by Jones et al. (2021) for UM-UKCA without and with nitrate. We find that the global change in AOD for the $LNO_x$ range considered (0–5.2 Tg N yr$^{-1}$) is ~ 0.0015, which is ten times smaller than the increase of ~ 0.0154 in AOD when nitrate is included (see our Section 4.5 and Figure 13). AOD is a bulk aerosol quantity, and we

believe that its measurements are unlikely to discern the magnitude of changes in the modelled AOD in response to changes in $LNO_x$ and nitrate, particularly $LNO_x$. Indeed, for example, Levy et al. (2013, https://doi.org/10.5194/amt-6-2989-2013) note that the uncertainty in MODIS global AOD cannot be reduced below ± 0.03, or 15–20% of global mean AOD. This uncertainty in observations is much greater than the above modelled variation in AOD in response to changes in $LNO_x$ and is also larger than the increase in AOD when nitrate is included. Therefore, an evaluation using AOD measurements would prove indecisive as the modelled AOD differences will most likely be within the observational uncertainty.

Essentially, to do a targeted evaluation of the impact of $LNO_x$ on aerosol, we need new data, e.g. from aircraft measurements in the tropics, coupled with some high-resolution regional chemical transport modelling, which is beyond the scope of the present study.

We agree with the Referee that the lightning chemistry-climate interactions via nitrate and cloud microphysics are very interesting. We believe more research is needed in this area, but for the present we include the following plot (which can go in either section 4.5 as Figure 15 or Supplement S2 where CDNC is discussed in more detail) on the impact of nitrate and $LNO_x$ on the modelled global-mean cloud droplet number concentration (CDNC). We state "We also examined the impact of nitrate on the modelled tropospheric cloud droplet number concentration (CDNC) (see Supplement S2) and found that compared to the no-LNOx case, there is a 3.1% increase in the mean tropospheric CDNC when $LNO_x$ is considered (via the Lu21 scheme). The incorporation of nitrate in the model causes an average increase of ~ 3.5% in CDNC for any given $LNO_x$ value. The average increase in CDNC per Tg N $yr^{-1}$ of $LNO_x$ is 0.035 $cm^{-3}$.

[Figure]

**Figure 15: Modelled annual-mean tropospheric cloud droplet number concentration (CDNC), as a function of lightning-generated NOx. The lines are linear least squares fits. The error bars correspond to a 1-sigma standard deviation.**